# What's Left After Distillation?
# How Knowledge Transfer Impacts Fairness and Bias

**Aida Mohammadshahi**                                              *aida.mohammadshahi@ucalgary.ca*

**Yani Ioannou**                                                    *yani.ioannou@ucalgary.ca*
*Department of Electrical and Software Engineering*
*Schulich School of Engineering, University of Calgary*
*Calgary, AB, Canada*

**Reviewed on OpenReview:** *https://openreview.net/forum?id=xBbj46Y2fN*

## Abstract

Knowledge Distillation is a commonly used Deep Neural Network (DNN) compression method, which often maintains overall generalization performance. However, we show that even for balanced image classification datasets, such as CIFAR-100, Tiny ImageNet and ImageNet, as many as 41% of the classes are statistically significantly affected by distillation when comparing class-wise accuracy (i.e. class bias) between a teacher/distilled student or distilled student/non-distilled student model. Changes in class bias are not necessarily an undesirable outcome when considered outside of the context of a model's usage. Using two common fairness metrics, Demographic Parity Difference (DPD) and Equalized Odds Difference (EOD) on models trained with the CelebA, Trifeature, and HateXplain datasets, our results suggest that increasing the distillation temperature improves the distilled student model's fairness, and the distilled student fairness can even surpass the fairness of the teacher model at high temperatures. Additionally, we examine individual fairness, ensuring similar instances receive similar predictions. Our results confirm that higher temperatures also improve the distilled student model's individual fairness. This study highlights the uneven effects of distillation on certain classes and its potentially significant role in fairness, emphasizing that caution is warranted when using distilled models for sensitive application domains.

## 1 Introduction

DNNs require significant computational resources, resulting in large overheads in compute, memory, and energy. Decreasing this computational overhead is necessary for many real-world applications where these costs would otherwise be prohibitive, or even make their application infeasible — e.g. the deployment of DNNs on mobile phones or edge devices with limited resources (Chen et al., 2016; Cheng et al., 2018; Gupta and Agrawal, 2022; Menghani, 2023). To address this challenge, DNN model compression methods have been developed that reduce the size and complexity of DNNs while maintaining their generalization performance (Cheng et al., 2017). One such widely used model compression method is Knowledge Distillation (distillation) (Hinton et al., 2015). Distillation has found extensive application in both industry and academia across various domains of artificial intelligence, encompassing areas such as Natural Language Processing (NLP) (Jiao et al., 2019; Fu et al., 2021; Liu et al., 2020), speech recognition (Ng et al., 2018; Gao et al., 2019; Perez et al., 2020), and visual recognition (Yan et al., 2019; Dou et al., 2020; Chawla et al., 2021), specifically image classification (Zhu et al., 2019; Chen et al., 2019; Gou et al., 2021).

Distillation involves transferring knowledge from a complex model with superior performance (referred to as the *teacher*) to a simpler model (known as the *student*). In practice this allows the student model to achieve comparable or even better generalization than the teacher model, while using far fewer parameters (Hinton et al., 2015; Gou et al., 2021). Despite the widespread use of distillation, evaluation of the impact of distillation since its proposal by (Hinton et al., 2015) has overwhelmingly focused almost exclusively on the impact it has on generalization performance (Cho and Hariharan, 2019; Mirzadeh et al., 2020).

Hooker et al. (2019) recently demonstrated however, that test accuracy alone was not sufficient to understand the full impact of two other popular approaches to reducing DNN complexity: *pruning* and *quantization*. They identified that even when pruning maintains the overall test accuracy of a model, it can affect the algorithmic bias of the pruned model.

Does the application of distillation uniformly affect accuracy across all classes, or are certain classes more significantly affected by distillation as with pruning? What is the impact of distillation on the model's bias and fairness? These questions are increasingly vital given the growing usage of distilled models (the models trained using distillation) in applications making impactful decisions. Even when the teacher model itself has been through a bias or fairness analysis, any changes in the biases of the student model due to distillation may lead to unforeseen consequences, potentially resulting in harmful outcomes for individuals and society as a whole.

In this work, we propose a structured empirical approach to evaluate the impact of distillation, to thoroughly examine if distillation significantly influences a distilled model's bias and class-wise accuracies for balanced datasets, and if the model's fairness changes for datasets with demographic attributes. To the best of our knowledge, our study is the first that evaluates the impact of temperature in distillation on both model bias and fairness, revealing important novel insights including:

- The impact of distillation is not random: there is a statistically significant difference in class-level accuracy between a non-distilled/distilled student, and teacher/distilled student.
- The number of significantly affected classes increases with higher temperatures comparing non-distilled vs. distilled student, whereas with teacher vs. distilled student they decrease.
- Distillation influences the class bias and fairness of distilled student models, even where there is no substantial change in the model's overall test accuracy.
- Distillation improves the student model's fairness compared to the non-distilled student model concerning demographic attributes, and employing higher temperatures increases the model's fairness (this does not hold for extreme temperatures like 20,30, and 40).
- Enhancements in fairness may not always align with improvements in the model's generalization performance; a trade-off should be selected based on the specific application.

## 2   Background

**Algorithmic Bias & Fairness**   The systemic bias that can be present in the decisions and predictions made by Machine Learning (ML) models is called algorithmic bias (Mehrabi et al., 2021). The analysis of algorithmic bias in training ML models has been a focus of many recent works (Barocas and Selbst, 2016; Mehrabi et al., 2021; Suresh and Guttag, 2021; Angwin et al., 2022). There are a wide range of sources of bias in the deep learning context, although existing bias in the training dataset is one of the main reasons for biased prediction (Hellström et al., 2020). For example, most real-world natural image datasets exhibit a notable long-tail distribution, characterized by varying frequencies of attributes in the dataset, with some attributes being significantly more prevalent than others. This under-representation of certain groups in the dataset could lead to biased model behaviour. The distribution of attributes in the CelebA dataset can be seen in Appendix A, Figure 6, as an example. The bias present in the training dataset may originate from inherited historical bias during data collection, the process of sampling the dataset, and the measurement of features and labels (Suresh and Guttag, 2021). Many works study data bias (Khan and Fu, 2021), for example Leevy et al. (2018) study the effect of class imbalance on learning, while Wang et al. (2020) study the effect of data bias on visual recognition tasks.

**Algorithmic Bias in Models/Methods**   However, bias is not merely a data problem, and the model's architecture and the algorithms used for learning can play an important role in increasing or curbing algorithmic bias (Hooker, 2021; Suresh and Guttag, 2021; Angwin et al., 2022). Zhang et al. (2018) in particular highlighted that certain neural network structures or learning paradigms can inherently amplify pre-existing biases in data.

**Fairness Metrics**   Bias in a model can lead to unfair outcomes. When a model consistently favours certain groups over others due to its biases, it fails to treat individuals equitably, resulting in unfair decisions (Pessach and Shmueli, 2022). Fairness metrics in ML are quantitative measures used to assess whether a model's predictions are fair, w.r.t a formal definition of fairness, particularly with respect to different demographic

groups. These metrics help in identifying biases and ensuring that the model treats all groups equitably. There are many fairness metrics, and they are often application-specific. In our work we will focus on two of the most commonly used: demographic parity (Calders et al., 2009) and equalized odds (Hardt et al., 2016).

**Demographic Parity**   Demographic Parity is achieved when the probability of a positive prediction $\widehat{Y}$ (e.g. predicting a common facial attribute, such as smiling or wearing sunglasses) is the same across different demographic groups $A$ (e.g. different genders, races, or age groups). For $\forall a,b \in A$:

$$P(\widehat{Y}=1\,|\,A=a)=P(\widehat{Y}=1\,|\,A=b).$$

**Equalized Odds**   Equalized Odds is achieved when both the true positive rate (the rate at which the model correctly identifies the positive class) and the false positive rate (the rate at which the model incorrectly identifies the negative class as positive) are similar across different demographic groups. For $\forall a,b \in A, y \in \{0,1\}$:

$$P(\widehat{Y}=1\,|\,Y=y,A=a)=P(\widehat{Y}=1\,|\,Y=y,A=b).$$

For comparing these metrics between different models, we use the DPD and EOD defined using the True Positive Rate (TPR) and False Positive Rate (FPR) to have a more nuanced understanding of how closely each model approaches fairness (Jung et al., 2021). These metrics evaluate how far a given predictor departs from satisfying a fairness constraint. The model with the lower difference value is closer to achieving these metrics and a value of zero signifies complete fairness from the perspective of that metric. DPD and EOD are calculated as follows:

$$\text{DPD} = \max_{a \in A} P(\widehat{Y}=1\,|\,A=a) - \min_{a \in A} P(\widehat{Y}=1\,|\,A=a), \tag{1}$$

$$\text{TPR Difference} = \max_{a \in A} P(\widehat{Y}=1\,|\,Y=1,A=a) - \min_{a \in A} P(\widehat{Y}=1\,|\,Y=1,A=a), \tag{2}$$

$$\text{FPR Difference} = \max_{a \in A} P(\widehat{Y}=1\,|\,Y=0,A=a) - \min_{a \in A} P(\widehat{Y}=1\,|\,Y=0,A=a), \tag{3}$$

$$\text{EOD} = \max(\text{TPR Difference}, \text{FPR Difference}). \tag{4}$$

In cases where the base rates of different demographic groups differ significantly, the Demographic Parity Difference (DPD) metric may not fully reflect fairness. Therefore, we recommend caution when interpreting DPD, and note that we also consider other fairness metrics, such as Equalized Odds Difference (EOD).

**Individual Fairness**   In addition, we extend our work to analyzing individual fairness, ensuring that similar individuals receive similar predictions. To evaluate this, we employ the Lipschitz condition proposed by Dwork et al. (2012), which quantifies fairness as the degree of variation in predictions with respect to input similarity:

$$\mathcal{I}(f) = \mathbb{E}_{(x,x') \sim P} \left[ \frac{|f(x)-f(x')|}{d(x,x')} \right], \tag{5}$$

where $d(x,x')$ measures the similarity between inputs using cosine similarity in a pre-trained feature space. This metric captures whether a model provides consistent predictions for semantically similar inputs, ensuring fairness at an individual level. By incorporating this analysis, we present a more granular and interpretable view of fairness in knowledge distillation.

**Model Compression Methods**   Given the high costs in deploying DNN models for many applications, a general class of methods to address this problem are those of *model compression*. Model compression techniques are commonly employed in DNN applications because they can drastically reduce the computational and/or memory footprint of a DNN model, with little effect on generalization performance. These methods include pruning (Han et al., 2015b; Zhu and Gupta, 2017a; Luo et al., 2017), which entails discarding insignificant or redundant model parameters, quantization (Gong et al., 2014a; Han et al., 2015a; Mishra et al., 2017), which involves lowering the precision of the model's weights and activations, and distillation (Hinton et al., 2015), which trains a smaller model (the student model) to acquire knowledge from a more complex model (the teacher model).

**Knowledge Distillation**   Consider a large model that learns to classify between a large number of classes in an image dataset. This model takes any input images and produces logits, $z_i$, raw scores that are associated with each class $i$, as the outputs of the last layer of the model. It then uses the softmax function that takes the logits and transforms them into probabilities, $p_i$, and ultimately chooses the class with the highest probability as its prediction. In distillation the softmax equation is redefined:

$$p_i = \frac{e^{z_i/T}}{\sum_j e^{z_j/T}}, \tag{6}$$

where $T$ is the *temperature* hyper-parameter. The probabilities of incorrect classes can provide valuable insights into how the large model generalizes. However, these probabilities are often very small and close to zero when $T=1$. By increasing the value of the temperature $T$ in Equation (6), we will have a softer probability distribution over classes (known as *soft targets*) with higher entropy, that represent a richer representation for distillation training than *hard targets* (ground truth labels). In distillation, we train a smaller model (student) to use the soft targets generated by the pre-trained larger model (teacher) along with the hard targets to learn how to generalize the same as the larger model (Hinton et al., 2015). The student model uses a weighted sum of two different objective functions: The distillation loss, a cross-entropy with soft targets, which is computed using the same high temperature used in the softmax of the larger model to generate the soft targets, and the classification loss, a cross-entropy with the actual labels:

$$L_{total} = \alpha \times L_{Distillation} + (1-\alpha) \times L_{Classification}. \tag{7}$$

## 3   Related Work

**Bias in Model Compression Methods**   Considering the significant expenses associated with deploying DNN models across various applications, a category of methods which have to date received less attention vis-a-vis algorithmic bias are those of *model compression*. The studies of model compression methods are overwhelmingly focused on demonstrating that such methods can significantly decrease the computational and/or memory footprints of a DNN model while having minimal impact on generalization performance (Gong et al., 2014b; Han et al., 2015a; Zhu and Gupta, 2017b; Luo and Wu, 2017; Wang et al., 2019).

While model compression methods often maintain generalization performance, they may do so by learning different functions than that learned in the uncompressed model, and hence exhibit different algorithmic bias. Hooker et al. (2019) were amongst the first to emphasize this research direction, focusing on the implications of pruning and quantization on a model's algorithmic bias, and identifying both of these methods could exacerbate the algorithmic bias of a model. Joseph et al. (2020) also studied the misalignments in predictions and fairness metrics of uncompressed and compressed models using group sparsity and showed that certain classes experience more pronounced impacts than others.

Hooker et al. (2020) continued their previous work and leveraged a subset of data points to investigate the implications of pruning and post-training quantization on model bias and fairness considerations. They found that compression consistently exacerbates the unequal treatment of underrepresented protected subgroups for all levels of compression, and demonstrated that there exists a set of data carrying a disproportionately significant share of the error. Later, Stoychev and Gunes (2022) extended these studies to the domain of facial expression recognition and confirmed that compressing models can amplify pre-existing biases related to gender in models trained for facial expression recognition.

Despite its prevalence, fewer studies have been conducted on understanding the impact on bias and fairness of distillation. Lukasik et al. (2021) showed that improvements in overall accuracy by self-distillation may come at the expense of harming accuracy on certain classes, and in a study by Chai et al. (2022), the influence of the weight $\alpha$ in the distillation loss (Equation (7)) on model fairness was analyzed, revealing that an increase in $\alpha$ enhances fairness. In our research, we broaden the investigation into distillation, assessing the effects of increasing the temperature parameter on class-level performance and fairness in the student model.

## 4 Methodology

**Metrics and Matrices** We want to understand if the performance of the model across various classes and groups remains unchanged when using distillation and if any changes occur, whether they are statistically significant. While Hinton et al. (2015) originally motivated distillation in the context of distilling a teacher as an ensemble of models to a single student model, distilling a single large teacher model to a smaller student model more accurately matches the most common contemporary use case for distillation. Similarly, we train a large complex model as a teacher, and then use this pre-trained teacher to train a smaller student model using distillation with the chosen temperature *(distilled student)*. As a baseline we also train the student model *(the non-distilled student)* separately as normal, i.e. from random initialization with just the hard targets (class labels). In addition to standard metrics such as class-wise accuracy, we define a matrix of disagreement to compare the predictions of the teacher and the distilled student model, as well as the teacher and the non-distilled student. This matrix provides valuable information about the number of disagreements between models. The definition of disagreement follows Fort et al. (2020):

$$CMP(f(x_n), g(x_n)) = \begin{cases} 0 & \text{if} \quad f(x_n) = g(x_n) \\ 1 & \text{if} \quad f(x_n) \neq g(x_n). \end{cases} \tag{8}$$

Here, CMP denotes the Comparison of Model Predictions. If the models disagree on the prediction for a given instance, a value of 1 is recorded for that instance's class.

To understand the significance of our results, we carry out five separate training runs with different random initializations for each combination of teacher and student models, and calculate the average of their respective disagreement matrices.

**Impact measurement** To measure the significance of the impact of distillation on bias, we use a two-tailed Welch's t-test (Welch, 1947), which is a statistical method commonly used in hypothesis testing. Assuming that the effect of using distillation to train a student model is uniform, the correlation between the test-set accuracy at the class level and the overall model test accuracy over all classes must remain unchanged, i.e. the null hypothesis ($H_0$). The alternate hypothesis ($H_1$) states that the relative difference in class-wise accuracy is not equivalent to the change in overall accuracy. We need to determine which hypothesis to accept for each class $c_i$ at each temperature $T$. In our notation, $\beta^M$ represents the model's test accuracy over all classes and $\beta^{c_i}$ denotes class-wise test accuracy. When using distillation, we use $\beta_T^M$ and $\beta_T^{c_i}$ to indicate these accuracies at a specific distillation temperature $T$:

$$H_0 : \frac{\beta^{c_i}}{\beta^M} = \frac{\beta_T^{c_i}}{\beta_T^M}, \qquad H_1 : \frac{\beta^{c_i}}{\beta^M} \neq \frac{\beta_T^{c_i}}{\beta_T^M}. \tag{9}$$

Our goal is to have a high level of certainty that changes in the relative class-wise accuracy of the distilled model are statistically significant and not due to inherent noise in the stochastic training process of DNN. To achieve this goal, for each combination of temperature, dataset, and student-teacher networks that we choose, we train a group of five teacher and five student models independently with different initializations. Consequently, we obtain a set of accuracy metrics per class $c$ for teacher, non-distilled student, and distilled students. We evaluate whether there are significant differences between the class accuracy of distilled and non-distilled student, as well as distilled student and teacher sets normalized by their respective overall model accuracy. If the p-value is less than or equal to 0.05 for each class, the null hypothesis is rejected and we conclude that training the student model with distillation has a significant effect on the relative class performance.

If a uniform drop occurs across all classes, the relative change in class-wise accuracy would remain consistent with the overall accuracy drop. In such a case, we would not reject the null hypothesis because the changes are not class-specific or biased, and there would be no evidence that distillation has a disproportionately larger effect on certain classes. However, if the accuracy drop is not uniform — i.e., some classes experience a larger drop in accuracy than others — this would indicate that distillation affects certain classes differently, and the relative

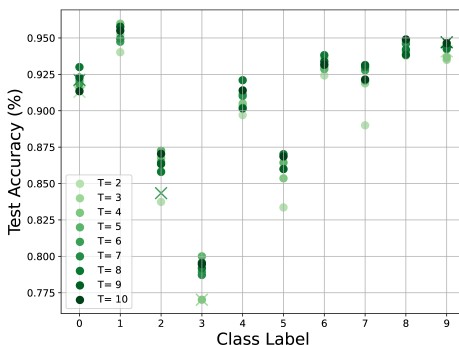 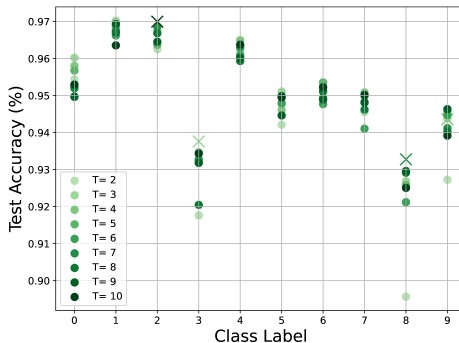

Figure 1: **Class-wise Bias and Distillation.** Test Accuracies of ResNet-20 student models distilled from a ResNet-56 teacher on CIFAR-10 (left) and SVHN (right) over a range of temperatures $T$. Mean test accuracies are shown over five random initializations. Classes with statistically significant relative changes between the non-distilled student and the distilled student are noted with $\times$.

Table 1: **Class-wise Bias and Distillation.** The number of statistically significantly affected classes comparing the class-wise accuracy of *teacher vs. Distilled Student (DS) models*, denoted #TC, and *Non-Distilled Student (NDS) vs. distilled student models*, denoted #SC.

| | | CIFAR-100 | | | | | | ImageNet | | | | | |
| | | ResNet56/ResNet20 | | | DenseNet169/DenseNet121 | | | ResNet50/ResNet18 | | | ViT-Base/TinyViT | | |
| Teacher/Student | | | | | | | | | | | | | |
| Model | Temp | Test Acc. (%) | #SC | #TC | Test Acc. (%) | #SC | #TC | Test Top-1 Acc. (%) | #SC | #TC | Test Top-1 Acc. (%) | #SC | #TC |
|---|---|---|---|---|---|---|---|---|---|---|---|---|---|
| Teacher | - | $70.87 \pm 0.21$ | - | - | $72.43 \pm 0.15$ | - | - | $76.1 \pm 0.13$ | - | - | $81.02 \pm 0.07$ | - | - |
| NDS | - | $68.39 \pm 0.17$ | - | - | $70.17 \pm 0.16$ | - | - | $68.64 \pm 0.21$ | - | - | $78.68 \pm 0.19$ | - | - |
| DS | 2 | $68.63 \pm 0.24$ | 5 | 15 | $70.93 \pm 0.21$ | 4 | 12 | $68.93 \pm 0.23$ | 77 | 314 | $78.79 \pm 0.21$ | 83 | 397 |
| DS | 3 | $68.92 \pm 0.21$ | 7 | 12 | $71.08 \pm 0.17$ | 4 | 11 | $69.12 \pm 0.18$ | 113 | 265 | $78.94 \pm 0.14$ | 137 | 318 |
| DS | 4 | $69.18 \pm 0.19$ | 8 | 9 | $71.16 \pm 0.23$ | 5 | 9 | $69.57 \pm 0.26$ | 169 | 237 | $79.12 \pm 0.23$ | 186 | 253 |
| DS | 5 | $69.77 \pm 0.22$ | 9 | 8 | $71.42 \pm 0.18$ | 8 | 9 | $69.85 \pm 0.19$ | 190 | 218 | $79.51 \pm 0.17$ | 215 | 206 |
| DS | 6 | $69.81 \pm 0.15$ | 9 | 8 | $71.39 \pm 0.22$ | 8 | 8 | $69.71 \pm 0.13$ | 212 | 193 | $80.03 \pm 0.19$ | 268 | 184 |
| DS | 7 | $69.38 \pm 0.18$ | 10 | 6 | $71.34 \pm 0.16$ | 9 | 7 | $70.05 \pm 0.18$ | 295 | 174 | $79.62 \pm 0.23$ | 329 | 161 |
| DS | 8 | $69.12 \pm 0.21$ | 13 | 6 | $71.29 \pm 0.13$ | 11 | 7 | $70.28 \pm 0.27$ | 346 | 138 | $79.93 \pm 0.12$ | 365 | 127 |
| DS | 9 | $69.35 \pm 0.27$ | 18 | 9 | $71.51 \pm 0.23$ | 12 | 9 | $70.52 \pm 0.09$ | 371 | 101 | $80.16 \pm 0.17$ | 397 | 96 |
| DS | 10 | $69.24 \pm 0.19$ | 22 | 11 | $71.16 \pm 0.21$ | 14 | 10 | $70.83 \pm 0.15$ | 408 | 86 | $79.98 \pm 0.12$ | 426 | 78 |

difference between class-wise accuracy and overall accuracy would not be consistent. In such a case, we would reject the null hypothesis and conclude that the distillation process significantly impacts class-level accuracy.

## 5 Experimental Design and Results

**Class-wise Measurement** In our experiments, we explore a variety of teacher/student model architectures across multiple datasets. We use the SVHN (Netzer et al., 2011), CIFAR-10 (Krizhevsky, 2009), CIFAR-100 (Krizhevsky, 2009), Tiny ImageNet (Le and Yang, 2015), and ImageNet (ILSVRC 2012) (Deng et al., 2009) datasets, all recognized benchmarks image classification. Further details on these datasets are given in Appendix A. All these datasets are balanced with an almost equal number of instances across their classes. Due to their balanced nature, these datasets don't demonstrate the long-tailed imbalanced distribution that many real-world datasets have, and if anything models trained on these datasets should be less susceptible to dataset bias.

To evaluate the effect of using distillation, we distil knowledge from a teacher ResNet-56 (He et al., 2016) model to train a student ResNet-20 model on the SVHN and CIFAR-10 datasets, for CIFAR-100 we evaluate ResNet-56/ResNet-20 and DensNet-169/DenseNet-121 (Huang et al., 2017) teacher/student models, for Tiny Imagenet ResNet-50/ResNet-18, and for ImageNet, we evaluate ResNet-50/ResNet-18 and ViT-Base/TinyViT (Dosovitskiy et al., 2020; Wu et al., 2022) teacher/student models. We use $\alpha = 0.8$ for all experiments to have consistency. The experiments with other $\alpha$ show the same trend as shown in Appendix B, Table 8.

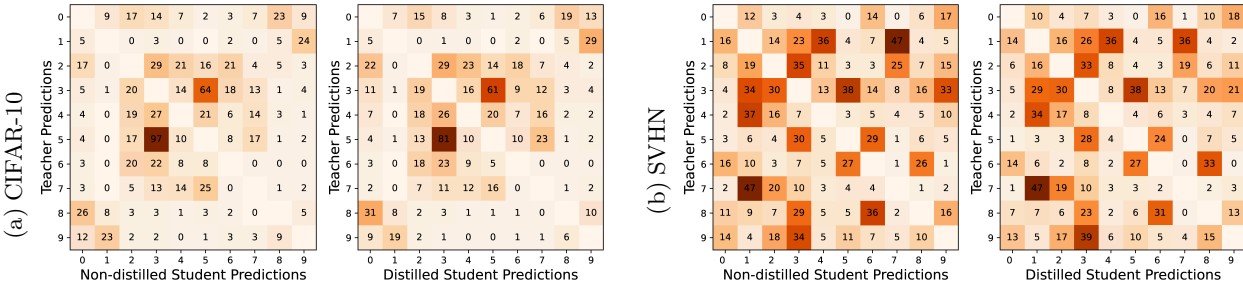

Figure 2: **Class-wise Disagreement.** Disagreement between a ResNet-56 teacher and ResNet-20 (left) non-distilled/(right) distilled student for (a) CIFAR-10 using $T = 9$ and (b) SVHN using $T = 7$. The diagonals are excluded since here both models predict the same class without any disagreement.

The baseline results for teacher models are listed in Tables 1 and 6. The baseline *non-distilled* student models achieve lower test accuracies compared to their corresponding teacher models, as also listed in the tables, as is typically observed for smaller models empirically. Full experimental details can be found in Appendix C.

**Temperature** During normal model training, temperature $T = 1$ is used for the softmax (Equation (6)). However, in distillation, we use a higher value of $T$ to create a softer probability distribution over classes in the pre-trained teacher network, which can then be used to train the student. We experiment with $T = 2$–$10$, a common range for T found in distillation literature (Hinton et al., 2015; Cho and Hariharan, 2019; Stanton et al., 2021) and expand the results to very high temperatures, $T = \{20, 30, 40\}$, in the Appendix.

In Figure 1 and Table 6, we observe the changes in class-level accuracies of student models that have undergone distillation at varying temperatures, trained on the CIFAR-10 and SVHN datasets. The lowest accuracies across most classes are noted at T=2, suggesting that the knowledge transferred from the teacher network at this temperature carries less informative content compared to higher temperatures. Notably, at T=2, 3, 6, 7, and 9, one class is significantly affected comparing the non-distilled student and the distilled student for CIFAR-10. Similarly, for SVHN, significant changes in a single class are seen at T=3, 4, 7, and 10. These findings indicate that the influence of distillation on class performance is not uniform for most temperatures. In addition, at certain temperatures, there is one significant class comparing the teacher and distilled student (Table 6).

Figure 2 illustrates the disagreement matrices comparing predictions between the teachers and the non-distilled students, as well as between the teachers and the distilled students. Differences in these matrices highlight the influence of distillation on the distilled student model. In these datasets, while the number of images per class is balanced, the complexity and variability within each class can differ. Some classes are inherently more challenging to classify due to factors like intra-class variation or similarity to other classes. In the case of CIFAR-10, notable disagreements arise particularly for the cat (labelled 3) and dog (labelled 5) classes, when comparing predictions of the teacher and non-distilled student models. These disagreements are reduced between the teacher and the distilled student as a result of distilling knowledge from the teacher model. For SVHN, where class complexities are more similar, the predominant disagreements are observed for digits 1 and 7. Similarly, we see a reduction in these disagreements following the application of distillation.

Figure 3 displays the number of classes in the CIFAR-100 and ImageNet datasets that demonstrate statistically significant relative changes. We note a direct relationship between the increase in the temperature parameter used in distillation and the rise in the count of non-distilled student vs. distilled student significantly affected classes. Furthermore, as the temperature increases, the count of teacher vs. distilled student significantly affected classes experiences a decline. This observation implies that as the temperature rises, increasing the entropy of the targets used to transfer knowledge from the teacher to the student, the distilled student aligns more closely with the biases of the teacher. Conversely, at lower temperatures, the distilled student tends to be closer to the non-distilled student trained from scratch. It is important to highlight that the observed significantly affected classes do not inherently indicate positive or negative outcomes, as the model accuracy demonstrates the potential for both improvement and decay. However, this pattern indicates a clear impact of distillation on the student model's bias across various classes. Additionally, a comparison across the datasets in Table 1 and Table 6 indicates a

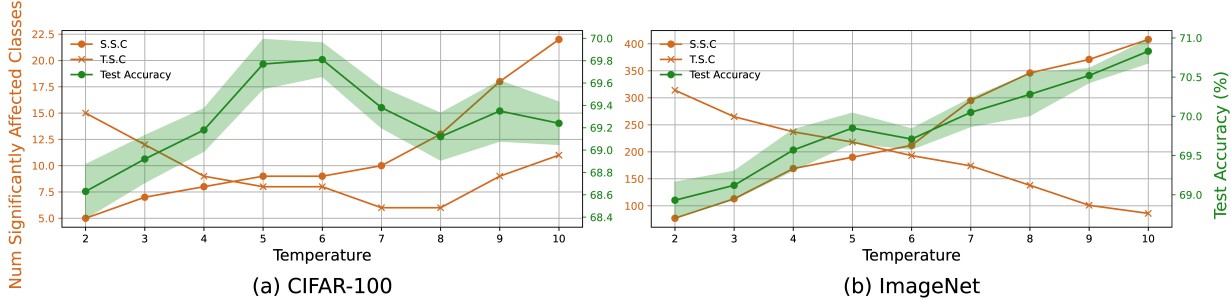

Figure 3: **Temperature vs. Test Accuracy/Class Bias.** Number of non-distilled vs. distilled student significantly affected classes (S.S.C.) and the number of teacher vs. distilled student significantly affected classes (T.S.C.) by distillation in (a) CIFAR-100 (ResNet-56/ResNet-20) and (b) ImageNet datasets (ResNet-50/ResNet-18), with 100 and 1000 total classes respectively. As the temperature used for distillation increases up to T=10, the S.S.C. rises for both datasets. For ImageNet, T.S.C. decreases, while for CIFAR-100, it first decreases and then slightly increases. The changes in the distilled student's test accuracy over all classes are also depicted in the figure.

correlation between dataset complexity and the proportion of classes significantly affected. In the case of SVHN and CIFAR-10, at most one class (out of ten) shows notable changes at varying temperatures. In contrast for the CIFAR-100, Tiny ImageNet and ImageNet datasets, a consistent presence of a significantly higher proportion of affected classes is observed at all temperatures, with a higher fraction of such classes in the ImageNet dataset, highlighting the nuanced effect of dataset complexity on the class-specific impacts of distillation.

Our findings reveal a consistent pattern across all datasets and distillation temperatures: a small subset of classes is disproportionately affected by distillation. This impact is not random and shows a statistically significant difference in class-level accuracy between non-distilled and distilled student models as well as between teacher and distilled student. Moreover, this effect becomes more pronounced at higher temperature levels, as an increasing number of classes show a significant relative change in accuracy between non-distilled and distilled student models and the number of classes significantly different between teacher and distilled student reduces.

Evaluations at very high temperatures (T=20,30,40), shown in Appendix B, Table 7, and Figure 8 indicate that these trends do not persist, as should be expected. At very high temperatures the probability distribution will be very close to uniform with little additional information for the student model to learn from.

**Fairness Measurement** Based on our earlier experiments, we understand that distillation influences the class bias. We caution that the experiments on CIFAR-100 and ImageNet do not directly concern fairness metrics since these datasets do not contain sensitive attributes. While temperature affects class-wise accuracy, the fairness metrics are not directly applicable in these cases, as these datasets are balanced and lack demographic attributes. To quantify this change in bias and its impact on the model's fairness, we examine additional datasets that possess demographic attributes, allowing us to measure their fairness metrics effectively. One of the datasets we use is CelebFaces Attributes dataset (CelebA) (Liu et al., 2015) which is a comprehensive face attributes dataset, comprising over 200K celebrity images, each annotated with 40 different attributes. For our analysis, we select one of these attributes to serve as a binary label, focusing on training our model to classify whether the celebrity is smiling (Extra experiments with eyeglasses label can be found in Appendix B, Table 9). The CelebA dataset is particularly suitable for this study as it includes two key attributes, 'Young' and 'Male', which we identify as protected demographic attributes. This selection makes CelebA an ideal choice for assessing the impact of distillation on fairness metrics in the context of these attributes.

The CelebA dataset, like many real-world natural image datasets, exhibits a long-tail distribution in its attributes, as shown in Appendix A, Figure 6. The dataset is imbalanced with respect to the smiling/not smiling label, with an unequal number of images in each category. To mitigate the effects of this imbalance w.r.t. smiling/not smiling on potential bias and enhance prediction accuracy, we randomly undersample the over-represented class, resulting in the distribution of training data shown in Table 5 across our label attribute and demographic groups after balancing the training data. Even though the quantity of instances labeled as smiling now matches the count of samples labeled as not smiling, there exists an imbalance within the demographic groups. For

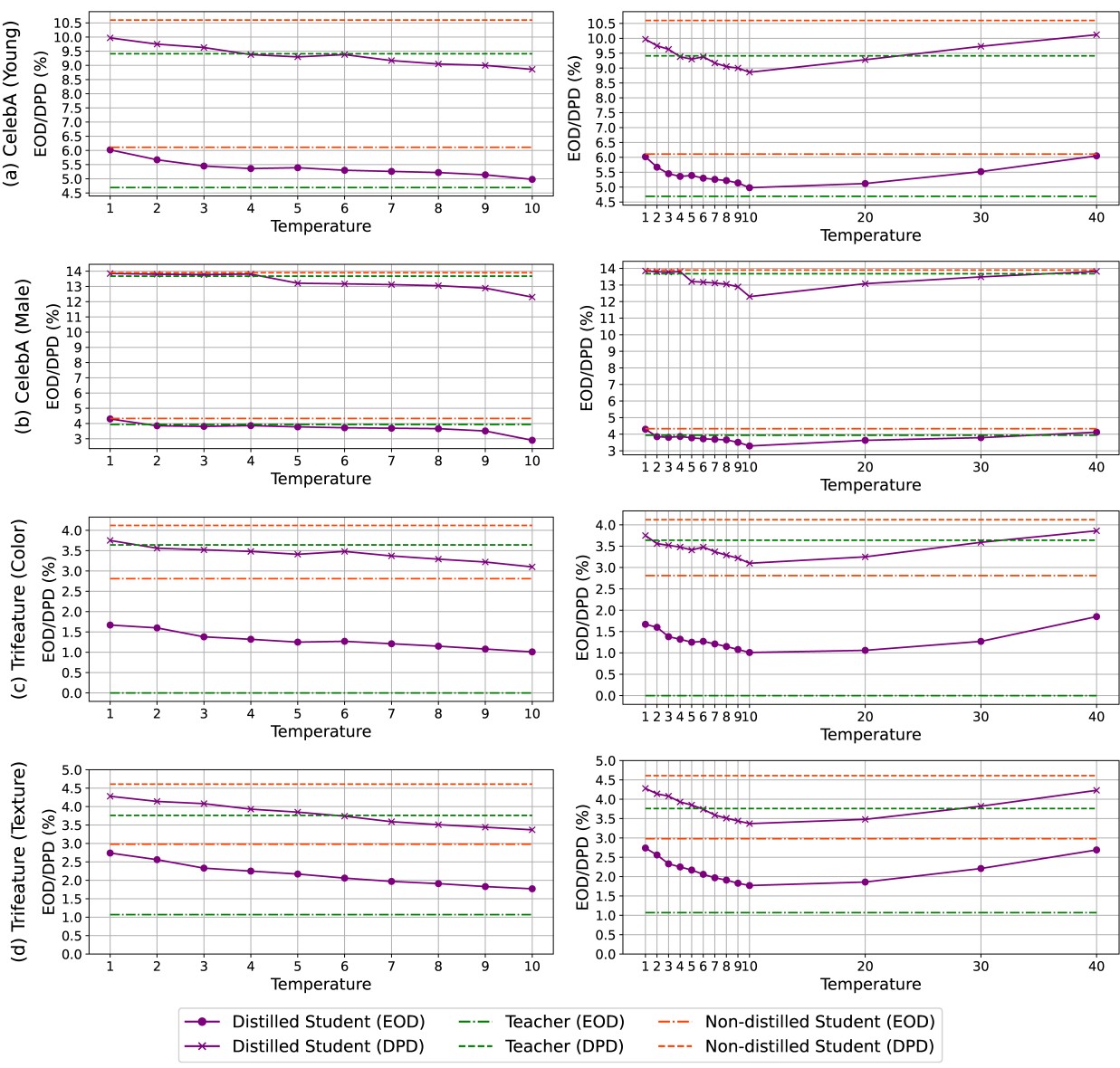

Figure 4: **Evaluation of Fairness Metrics for Distilled Students in Computer Vision (CV) .** Equalized Odds Difference (EOD) and Demographic Parity Difference (DPD) are reported in % and lower values indicate improved fairness. (a) illustrates fairness metrics for the CelebA dataset with 'smiling' label concerning the 'Young' demographic attribute and (b) concerning the 'Male' demographic attribute. (c) presents fairness metrics for the Trifeature dataset with 'shape' label with regard to the 'color' attribute and (d) with regard to the 'texture' attribute. It is notable that the models are fairer for the Trifeature dataset compared to the CelebA dataset with lower values in metrics. The explanation lies in the fact that the Trifeature dataset maintains a balanced distribution of demographic attributes, while the CelebA dataset contains biases that mirror real-world disparities. As seen in the second column, the downward trend does not continue at very high temperatures (T=20,30,40), as the teacher model generates nearly uniform softmax outputs.

Table 2: **Fairness Metrics and Distillation.** The performance of teacher, Non-Distilled Student (NDS), and Distilled Student (DS) models with a range of temperatures $T$ on the Trifeature and CelebA datasets. Fairness metrics are presented for Trifeature with regard to color attribute and for CelebA with regard to the Young demographic attribute. With increasing temperature, EOD and DPD have a downward trend signifying enhanced fairness. Mean and std. dev. are over five random inits.

| | | Trifeature (shape) | | | CelebA (smiling) | | |
| | | ResNet-20 / LeNet-5 | | | ResNet-50 / ResNet-18 | | |
| Model | Temp | Test Acc. (%) ↑ | EOD ↓ | DPD ↓ | Test Acc. (%) ↑ | EOD ↓ | DPD ↓ |
|---|---|---|---|---|---|---|---|
| Teacher | – | $100 \pm 0.00$ | $\mathbf{0.00 \pm 0.00}$ | $3.64 \pm 0.00$ | $93.09 \pm 0.08$ | $\mathbf{4.69 \pm 0.06}$ | $9.41 \pm 0.11$ |
| NDS | – | $90.33 \pm 0.07$ | $2.81 \pm 0.05$ | $4.12 \pm 0.08$ | $92.03 \pm 0.03$ | $6.11 \pm 0.05$ | $10.60 \pm 0.08$ |
| DS | 1 | $92.70 \pm 0.15$ | $1.67 \pm 0.04$ | $3.75 \pm 0.07$ | $92.12 \pm 0.06$ | $6.02 \pm 0.11$ | $9.97 \pm 0.08$ |
| DS | 2 | $92.41 \pm 0.08$ | $1.60 \pm 0.06$ | $3.56 \pm 0.05$ | $92.14 \pm 0.11$ | $5.67 \pm 0.08$ | $9.75 \pm 0.09$ |
| DS | 3 | $92.66 \pm 0.12$ | $1.38 \pm 0.02$ | $3.52 \pm 0.02$ | $92.53 \pm 0.13$ | $5.45 \pm 0.05$ | $9.63 \pm 0.06$ |
| DS | 4 | $92.61 \pm 0.17$ | $1.32 \pm 0.05$ | $3.48 \pm 0.03$ | $92.17 \pm 0.10$ | $5.36 \pm 0.02$ | $9.38 \pm 0.03$ |
| DS | 5 | $93.13 \pm 0.09$ | $1.25 \pm 0.00$ | $3.41 \pm 0.06$ | $92.29 \pm 0.05$ | $5.39 \pm 0.04$ | $9.30 \pm 0.05$ |
| DS | 6 | $93.25 \pm 0.12$ | $1.27 \pm 0.02$ | $3.48 \pm 0.03$ | $92.26 \pm 0.08$ | $5.30 \pm 0.01$ | $9.38 \pm 0.07$ |
| DS | 7 | $92.85 \pm 0.18$ | $1.21 \pm 0.03$ | $3.37 \pm 0.05$ | $92.12 \pm 0.08$ | $5.26 \pm 0.05$ | $9.17 \pm 0.10$ |
| DS | 8 | $92.66 \pm 0.13$ | $1.15 \pm 0.05$ | $3.29 \pm 0.06$ | $92.66 \pm 0.12$ | $5.22 \pm 0.02$ | $9.05 \pm 0.04$ |
| DS | 9 | $93.18 \pm 0.08$ | $1.08 \pm 0.03$ | $3.22 \pm 0.04$ | $93.18 \pm 0.15$ | $5.14 \pm 0.04$ | $9.01 \pm 0.08$ |
| DS | 10 | $92.77 \pm 0.16$ | $1.01 \pm 0.04$ | $\mathbf{3.10 \pm 0.02}$ | $92.57 \pm 0.11$ | $4.98 \pm 0.03$ | $\mathbf{8.86 \pm 0.04}$ |

---

**Algorithm 1** Fairness Metrics Calculation for Trifeature Dataset

**Input:** List of classes $C$, Demographic attribute $A$
$AggregateFairness \leftarrow 0$
**for** each class $c$ in $C$ **do**
  Create binary labels for class $c$:
  $Y_{binary} \leftarrow \{1 \text{ if } y = c, \text{ else } 0 \text{ for each } y \text{ in } Y\}$
  $F_c \leftarrow \text{FairnessMetric}(Y_{binary}, \widehat{Y}_{binary}, A)$
  $AggregateFairness \leftarrow AggregateFairness + F_c$
**end for**
$\overline{F} \leftarrow \frac{AggregateFairness}{\text{len}(C)}$
**return** Overall Fairness Metric $\overline{F}$

---

example, the demographic group "Not Young" is under-represented, constituting only 22.15% of the training dataset. Subsequently, we train a ResNet-50 model to classify the binary smiling attribute. This model is then used as a teacher to guide the training of a ResNet-18 model through distillation at various temperatures. Additionally, we train a ResNet-18 model from scratch to serve as a non-distilled student baseline.

We calculate the fairness metrics for baseline models and distilled student with different temperatures. We compare the performance across demographic groups against the non-distilled student model baseline. This baseline allows us to measure how distillation impacts the model's fairness by showing whether it exacerbates or mitigates fairness issues that already existed in the baseline model. In some contexts, a small change in fairness metrics could be considered substantial, especially in sensitive domains (e.g., hiring algorithms, loan approvals, or medical diagnoses), where even slight improvements in fairness can have a significant social or ethical impact. On the other hand, in less critical applications, this change might not carry the same weight. Therefore, it's important to interpret these changes based on the domain-specific impact. The outcomes are presented in Figure 4 and Table 2, where an increase in temperature is correlated with improved fairness, having smaller demographic parity and equalized odds differences. The teacher baseline demonstrates superior fairness compared to the student baseline across both demographic attributes. It is notable that the fairness of the distilled student can surpass that of the teacher model as seen in the changes in fairness metrics in Figure 4. This improvement suggests that the knowledge transferred from the teacher model aids the student model in paying greater attention to demographic attributes, thereby promoting more equitable treatment. This trend does not continue at very high temperatures ($T = 20, 30, 40$), as the teacher model generates nearly uniform softmax outputs (Figure 4, and Appendix B Table 9).

Table 3: **Fairness Metrics and Distillation for HateXplain.** The performance of Teacher, Non-Distilled Student (NDS), and Distilled Student (DS) models with varying temperatures $T$ on the HateXplain dataset. Fairness metrics are presented for gender and race demographic attributes. With increasing temperature, EOD and DPD exhibit a downward trend, indicating enhanced fairness. Mean and standard deviation are reported over five random initializations.

| Teacher/Student | | HateXplain (gender) | | | HateXplain (race) | |
| | | BERT-Base / DistilBERT | | | BERT-Base / DistilBERT | |
| Model | Temp | Test Acc. (%) ↑ | EOD ↓ | DPD ↓ | EOD ↓ | DPD ↓ |
| --- | --- | --- | --- | --- | --- | --- |
| Teacher | – | $75.79 \pm 0.19$ | $1.63 \pm 0.05$ | $1.10 \pm 0.04$ | $\mathbf{1.13 \pm 0.04}$ | $1.64 \pm 0.07$ |
| NDS | – | $73.28 \pm 0.12$ | $4.79 \pm 0.03$ | $2.87 \pm 0.05$ | $6.41 \pm 0.08$ | $3.16 \pm 0.05$ |
| DS | 1 | $73.60 \pm 0.25$ | $4.35 \pm 0.04$ | $2.74 \pm 0.05$ | $6.18 \pm 0.08$ | $3.02 \pm 0.03$ |
| DS | 2 | $74.44 \pm 0.21$ | $4.46 \pm 0.07$ | $1.84 \pm 0.03$ | $5.86 \pm 0.05$ | $2.46 \pm 0.04$ |
| DS | 3 | $73.76 \pm 0.19$ | $3.88 \pm 0.05$ | $1.66 \pm 0.08$ | $4.99 \pm 0.09$ | $2.19 \pm 0.06$ |
| DS | 4 | $74.05 \pm 0.22$ | $3.51 \pm 0.09$ | $1.45 \pm 0.09$ | $4.13 \pm 0.06$ | $2.03 \pm 0.09$ |
| DS | 5 | $74.12 \pm 0.17$ | $2.97 \pm 0.12$ | $1.23 \pm 0.07$ | $3.78 \pm 0.11$ | $1.97 \pm 0.05$ |
| DS | 6 | $74.25 \pm 0.21$ | $2.16 \pm 0.06$ | $1.16 \pm 0.04$ | $3.53 \pm 0.08$ | $1.71 \pm 0.09$ |
| DS | 7 | $74.58 \pm 0.24$ | $1.66 \pm 0.07$ | $1.09 \pm 0.06$ | $3.31 \pm 0.11$ | $1.54 \pm 0.08$ |
| DS | 8 | $74.31 \pm 0.27$ | $1.49 \pm 0.10$ | $0.98 \pm 0.07$ | $3.16 \pm 0.06$ | $1.21 \pm 0.05$ |
| DS | 9 | $73.73 \pm 0.16$ | $1.28 \pm 0.08$ | $0.76 \pm 0.10$ | $2.99 \pm 0.09$ | $1.16 \pm 0.03$ |
| DS | 10 | $74.27 \pm 0.24$ | $\mathbf{1.13 \pm 0.11}$ | $\mathbf{0.64 \pm 0.08}$ | $2.64 \pm 0.08$ | $\mathbf{1.04 \pm 0.08}$ |

We also apply our analysis to the Trifeature (Hermann and Lampinen, 2020) synthetic dataset, to further assess fairness metrics. This dataset comprises 224x224 RGB images, each featuring one of ten shapes, rendered in ten different textures and colored in ten hues. Example images from this dataset are shown in Appendix A, Figure 7. This dataset's balanced attribute representation eliminates the skewness typically seen in real-world data, enabling a focused study on the direct effects of distillation on model fairness and bias. In this study, we classify 'shape' as the label attribute, while 'color' and 'texture' serve as demographic attributes (Extra experiments with texture attribute can be found in Appendix B, Table 10). The explicit distinction among these attributes within the dataset offers a straightforward scenario for exploring the nuanced impact of distillation on fairness metrics. We utilize a ResNet-20 model as the teacher and a LeNet-5 (LeCun et al., 1998) model, adapted for color images, as the student. This training is conducted over a range of distillation temperatures. Additionally, a LeNet-5 model is trained from scratch as a non-distilled baseline for comparative analysis. Experimental details for CelebA and Trifeature can be found in Appendix C.

In this dataset, where our target has 10 classes, we initially calculate fairness metrics separately for each class, treating the other classes collectively as a single 'other' class, effectively creating a binary classification scenario for each analysis. This allows us to evaluate fairness metrics for each class against the aggregated rest, considering each of the demographic attributes. Subsequently, we aggregate these metrics to understand the overall fairness of the model by averaging the metrics across all classes. The pseudocode of our calculation is shown in Algorithm 1.

The findings from this dataset shown in Table 2 and Figure 4 are consistent with those from the CelebA dataset, demonstrating that an increase in the distillation temperature leads to enhancement in the student model's fairness. Furthermore, the findings reaffirm that the fairness of the distilled student model can surpass that of the teacher baseline. Just as we observed for class bias, this trend does not hold at very high temperatures (T= 20,30,40) with the teacher model having almost uniform softmax outputs (Figure 4, and Appendix B, Table 10).

In addition, we extend our analysis to the HateXplain text dataset (Mathew et al., 2021), a benchmark for hate speech, to demonstrate the generalizability of our findings across different modalities. HateXplain comprises posts gathered from Twitter and Gab, each annotated from three distinct viewpoints: a standard three-class classification (hate, offensive, or normal), the target community affected by the hate or offensive language in the post, and the specific rationales—segments of the text—that informed the annotation decisions.

For this study, we merge the offensive and hate speech categories into a single "toxic" class, simplifying the classification task to distinguishing between toxic (hate or offensive) and non-toxic (normal) posts. The dataset provides detailed annotations regarding sensitive groups (target communities). We aggregate these into broader

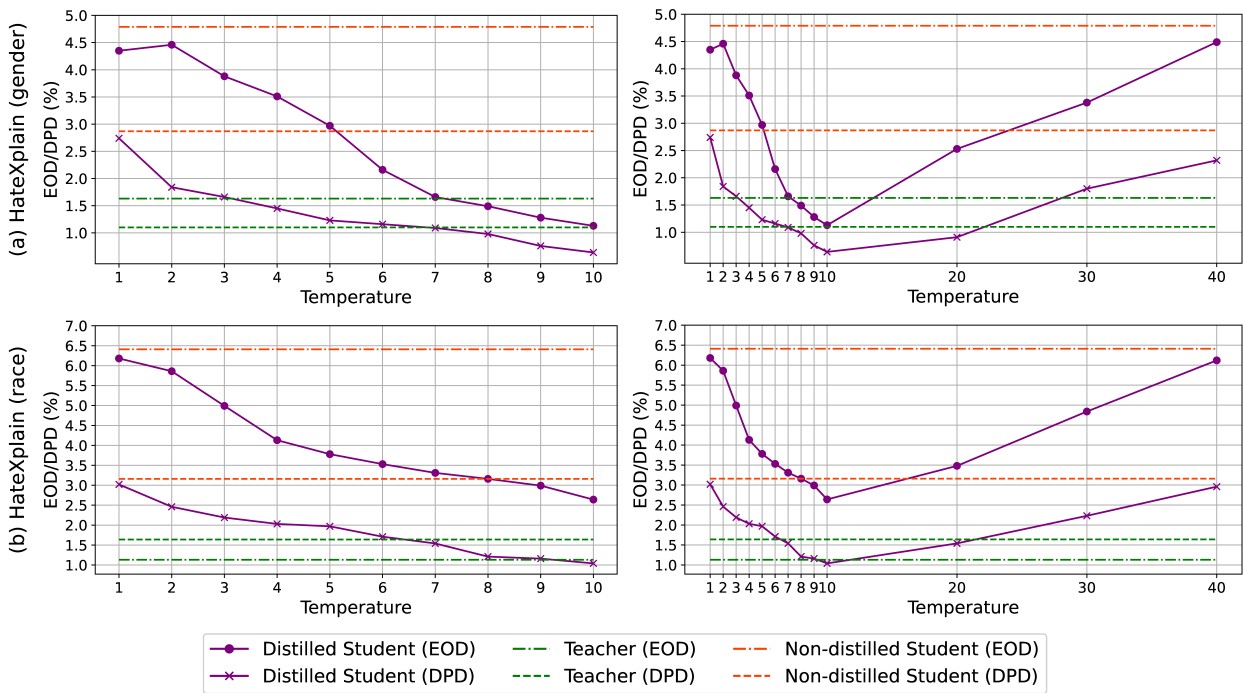

Figure 5: **Evaluation of Fairness Metrics for Distilled Students in Natural Language Processing (NLP).** Equalized Odds Difference (EOD) and Demographic Parity Difference (DPD) are reported in % and lower values indicate improved fairness. (a) illustrates fairness metrics for the HateXplain dataset concerning the 'gender' demographic attribute, and (b) with regard to the 'race' attribute. The teacher employed the BERT architecture, while the student used the DistilBERT architecture.

Table 4: **Individual Fairness Metrics Across Datasets.** Individual fairness scores for Teacher, Non-Distilled Student (NDS), and Distilled Student (DS) models across CelebA, Trifeature, and HateXplain datasets. Scores for DS models are reported for varying temperature values $T$.

| | | Individual Fairness ↓ | | |
| --- | --- | --- | --- | --- |
| | | **CelebA** | **Trifeature** | **HateXplain** |
| Model | Temp | ResNet-50 / ResNet-18 | ResNet-20 / LeNet-5 | BERT-Base / DistilBERT |
| Teacher | – | 0.0407 | 0.016 | 0.0320 |
| NDS | – | 0.124 | 0.0462 | 0.1078 |
| DS | 1 | 0.113 | 0.0422 | 0.0994 |
| DS | 2 | 0.104 | 0.0407 | 0.0985 |
| DS | 3 | 0.0908 | 0.0393 | 0.0927 |
| DS | 4 | 0.0906 | 0.0387 | 0.0882 |
| DS | 5 | 0.0886 | 0.0384 | 0.0823 |
| DS | 6 | 0.0799 | 0.0377 | 0.0768 |
| DS | 7 | 0.0753 | 0.0356 | 0.0727 |
| DS | 8 | 0.0712 | 0.0349 | 0.0689 |
| DS | 9 | 0.0701 | 0.0341 | 0.0681 |
| DS | 10 | **0.0697** | **0.0338** | **0.0654** |

categories by labeling a text as pertaining to a sensitive group if it references topics like race (e.g., African, Arab, Asian, Caucasian, Hispanic), and otherwise classifying it under the complementary group (no race).

We employ a BERT-Base (Devlin, 2018) model as the teacher model, guiding the training of a DistilBERT (Sanh, 2019) student model at various temperature settings. Additionally, we train a DistilBERT model independently from scratch to act as a non-distilled student baseline. The HateXplain dataset results presented in Table 3 and Figure 5 align with those from the CelebA and Trifeature datasets, confirming that increasing the KD temperature enhances the fairness of the student model even in NLP domain. In this experiment, we even see that both EOD and DPD surpass that of the teacher concerning the gender demographic attribute.

We also assess individual fairness across the CelebA, Trifeature, and HateXplain datasets, as shown in Table 4. The results indicate that as the temperature increases, the individual fairness of the distilled student model improves, with the best scores observed at higher temperature values. These findings suggest that distillation not only enhances fairness at the group level but can also improve individual fairness across various datasets. The individual fairness scores offer a more detailed view of how distillation affects the model's behavior toward specific instances, highlighting the potential of distillation to reduce bias in both group and individual decisions.

It is notable that with very large values of the temperature (T=20,30,40) and $\alpha$ distillation hyper-parameters, the student model does not receive as much useful information from the teacher. As the temperature increases, the teacher's soft labels become more uniform, reducing the distinctiveness of the information passed on to the student. This results in both a decrease in test accuracy and a negative impact on fairness metrics as seen in Table 9 and Table 10.

# 6 Discussion

Knowledge Distillation is an increasingly popular method for training DNNs, maintaining generalization performance while reducing the model's computational complexity. However, our findings reveal that distillation can otherwise affect the distilled model — specifically we found that even for nominally balanced benchmark image datasets, including CIFAR-10/100, SVHN, Tiny ImageNet and ImageNet, as many as 10–41% and 10–32% of classes are statistically significantly affected when comparing class-wise accuracy between a teacher/distilled student and distilled student/non-distilled student model respectively, varying based on the dataset and temperature (Table 1, Table 6). Our results (Figure 3) strongly suggest that the relationship between temperature and the number of significantly affected classes is characterized by larger temperatures maintaining the class biases found in the teacher model, while lower temperatures result in class biases closer to that of the baseline non-distilled student model. As already well understood, based on the specific dataset and/or model, there may be a significant difference in generalization performance with temperature — our results (Figure 3) suggest bias also be considered in this trade-off. We stress that changes in class bias are not necessarily an undesirable outcome in themselves, without an understanding of how that change affects decisions made in the context of a model's usage. Indeed our experimental results with two common fairness metrics, DPD and EOD, on three datasets with demographic groups: CelebA, Trifeature, and HateXplain suggest that by increasing the value of the temperature, a distilled student model's fairness improves. The distilled student fairness can even surpass the fairness of the teacher model at high temperatures (Figure 4).

Identifying challenges is a crucial step in advancing the field. While our results suggest a net positive impact of distillation on the distilled model's fairness, caution should be used before using distillation in sensitive domains. This study highlights the uneven effects of distillation on certain classes and its significant role in fairness. We are optimistic that our insights will inspire further advancements in the responsible application of distillation in practice.

**Acknowledgments**

We acknowledge the support of Alberta Innovates (ALLRP-577350-22, ALLRP-222301502), the Natural Sciences and Engineering Research Council of Canada (RGPIN-2022-03120, DGECR-2022-00358), and Defence Research and Development Canada (DGDND-2022-03120). This research was enabled in part by support provided by the Digital Research Alliance of Canada (`alliancecan.ca`). We are also grateful for computational resources made available to us by Denvr Dataworks and an Amazon Research Award. We also would like to acknowledge the helpful feedback of Sara Hooker, Mike Lasby, and Abbas Omidi.

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

## A Full Dataset Details

In our experiments, we explore a variety of teacher/student model architectures across multiple datasets. We use SVHN (Netzer et al., 2011), CIFAR-10, CIFAR-100 (Krizhevsky, 2009), Tiny ImageNet (Le and Yang, 2015) and ImageNet (ILSVRC) (Deng et al., 2009) datasets, all recognized benchmarks in real-world image classification.

The SVHN dataset, which includes house number images, is used in its cropped digits format, encompassing 10 classes for digits 0 through 9.

Both the CIFAR-10 and CIFAR-100 datasets comprise of 60,000 colour images of size 32x32, with CIFAR-10 divided into 10 classes, and CIFAR-100 divided into 100 classes, resulting in fewer samples per class in CIFAR-100.

Tiny ImageNet is a medium-scale image classification datasets categorized into 200 classes, containing 100,000 training and 10,000 test images from the ImageNet dataset, but downsized to $64 \times 64$ pixels. Each class has 500 training images, 50 validation images and 50 test images.

ImageNet is a large-scale image dataset categorized into 1,000 different classes, representing a diverse range of objects, animals, scenes, and concepts. All these datasets are balanced with an equal number of instances across their classes. Due to their balanced nature, these datasets don't demonstrate the long-tails imbalanced distribution that many real-world datasets have, and if anything models trained on these datasets should be less susceptible to dataset bias than otherwise.

## B  Extra Experiments

**Class-wise Bias**   Here we have included the full results tables for our class-wise bias analysis, these include CIFAR-10, SVHN and Tiny ImageNet in Table 6, CIFAR-100 and ImageNet in Table 7. We also show ablation results for other $\alpha$ hyperparameters for distillation in Table 8. Finally, we show the class-wise bias analysis including very higher temperatures (T>10) in Figure 8.

**Fairness**   Here we have included the full results tables for our fairness analysis using Celeb-A with both the "glasses" and "smiling" labels as attributes of interest in Table 9. We have also included the full results tables for the Tri-feature dataset using the "shape" and "texture" labels as attributes of interest in Table 10.

## C  Full Experimental Details

To evaluate the effect of using distillation, we distil knowledge from a teacher ResNet-56 (He et al., 2016) model to train a student ResNet-20 model on the SVHN and CIFAR-10 datasets, for CIFAR-100 we evaluate ResNet-56/ResNet-20 and DensNet-169/DenseNet-121 (Huang et al., 2017) teacher/student models, for Tiny Imagenet ResNet-50/ResNet-18, and for ImageNet, we evaluate ResNet-50/ResNet-18 and ViT-Base/TinyViT (Dosovitskiy et al., 2020; Wu et al., 2022) teacher/student models. We utilize data augmentation of random crop and horizontal flip, and adopt SGD optimizer with weight decay of 0.001, and decreasing learning rate schedules with a starting learning rate of 0.1 and a factor of 10 drop-off at epochs 30, 60, and 90 to train LeNet-5, ResNet and DenseNet models for 100 epochs. For training ViT models, we utilize the same settings mentioned in (Dosovitskiy et al., 2020), and (Wu et al., 2022). We train on SVHN, CIFAR-10, and CIFAR-100 with a batch size of 128 and on Tiny ImageNet and ImageNet with a batch size of 512

We trained LeNet-5, ResNet-56, ResNet-20, and DenseNet models using an NVIDIA RTX A4000 GPU with 16 GB of RAM. The training duration varied depending on the specific models and datasets used. For training the ResNet-50 and ResNet-18 models on Tiny ImageNet and ImageNet, we utilized four NVIDIA GeForce RTX 3090 GPUs, each with 24 GB of RAM. The training time was less than a day for Tiny ImageNet and approximately three days for ImageNet. For the ViT-Base and TinyViT models, we employed four NVIDIA A100 GPUs with 40 GB of RAM. The ViT-Base model took around two days to train, while TinyViT on Tiny ImageNet took less than a day.

For evaluating the effect of knowledge distillation on fairness, we train a ResNet-50 model on CelebA dataset as the teacher and use it to transfer knowledge to the ResNet-18 model as the student. We further experimented with Trifeature dataset and trained ResNet-20 as the teacher and LeNet-5, adapted for color images, as the student. Details of training these models are the same as previous trainings. We use the batch size of 256 for the CelebA dataset and batch size of 64 for Trifeature dataset.

Table 5: Distribution of training samples on the CelebA dataset balanced with respect to the smiling/not smiling attribute. The balance of the training dataset w.r.t. smiling/not smiling and age/gender attributes are listed vertically and horizontally respectively. (CelebA dataset does not account for non-binary gender)

|  | Age | | Gender | | |
|  | Young | Not Young | Male | Not Male | Distribution |
|---|---|---|---|---|---|
| Smiling | 61793 | 16287 | 37780 | 40300 | 50.0% |
| Not Smiling | 59765 | 18315 | 27259 | 50821 | 50.0% |
| Distribution | 77.85% | 22.15% | 41.65% | 58.35% | |

Table 6: **Class-wise Bias and Distillation.** The number of statistically significantly affected classes comparing the class-wise accuracy of *teacher vs. Distilled Student (DS) models*, denoted #TC, and *Non-Distilled Student (NDS) vs. distilled student models*, denoted #SC.

|  |  | CIFAR-10 | | | SVHN | | | Tiny ImageNet | | |
|  | Teacher/Student: | ResNet-56 / ResNet-20 | | | ResNet-50 / ResNet-18 | | | ResNet-50 / ResNet-18 | | |
| Model | Temp | Test Acc. (%) | #SC | #TC | Test Acc. (%) | #SC | #TC | Test Acc. (%) | #SC | #TC |
|---|---|---|---|---|---|---|---|---|---|---|
| Teacher | — | $92.18 \pm 0.12$ | — | — | $96.58 \pm 0.06$ | — | — | $72.3 \pm 0.07$ | — | — |
| NDS | — | $90.72 \pm 0.14$ | — | — | $95.31 \pm 0.08$ | — | — | $68.9 \pm 0.11$ | — | — |
| DS | 2 | $90.76 \pm 0.21$ | 1 | 1 | $95.34 \pm 0.16$ | 0 | 1 | $70.13 \pm 0.13$ | 16 | 32 |
| DS | 3 | $90.92 \pm 0.18$ | 1 | 1 | $95.53 \pm 0.12$ | 1 | 0 | $70.54 \pm 0.14$ | 19 | 28 |
| DS | 4 | $91.82 \pm 0.13$ | 0 | 0 | $95.80 \pm 0.02$ | 1 | 1 | $70.81 \pm 0.09$ | 21 | 23 |
| DS | 5 | $91.58 \pm 0.14$ | 0 | 0 | $95.77 \pm 0.06$ | 0 | 0 | $71.16 \pm 0.12$ | 24 | 20 |
| DS | 6 | $91.31 \pm 0.17$ | 1 | 1 | $95.89 \pm 0.13$ | 0 | 1 | $71.02 \pm 0.08$ | 28 | 17 |
| DS | 7 | $91.10 \pm 0.22$ | 1 | 0 | $96.10 \pm 0.08$ | 1 | 0 | $71.43 \pm 0.13$ | 33 | 15 |
| DS | 8 | $91.64 \pm 0.10$ | 0 | 0 | $95.76 \pm 0.14$ | 0 | 0 | $71.11 \pm 0.16$ | 39 | 14 |
| DS | 9 | $91.53 \pm 0.13$ | 1 | 0 | $95.83 \pm 0.11$ | 0 | 0 | $71.29 \pm 0.12$ | 43 | 12 |
| DS | 10 | $91.51 \pm 0.19$ | 0 | 1 | $95.86 \pm 0.15$ | 1 | 1 | $71.06 \pm 0.11$ | 49 | 10 |

Table 7: **Class-wise Bias and Distillation.** The number of statistically significantly affected classes comparing the class-wise accuracy of *teacher vs. Distilled Student (DS) models*, denoted #TC, and *Non-Distilled Student (NDS) vs. distilled student models*, denoted #SC.

|  |  | CIFAR-100 | | | | | | ImageNet | | | | | |
|  | Teacher/Student | ResNet56/ResNet20 | | | DenseNet169/DenseNet121 | | | ResNet50/ResNet18 | | | ViT-Base/TinyViT | | |
| Model | Temp | Test Acc. (%) | #SC | #TC | Test Acc. (%) | #SC | #TC | Test Top-1 Acc. (%) | #SC | #TC | Test Top-1 Acc. (%) | #SC | #TC |
|---|---|---|---|---|---|---|---|---|---|---|---|---|---|
| Teacher | - | $70.87 \pm 0.21$ | - | - | $72.43 \pm 0.15$ | - | - | $76.1 \pm 0.13$ | - | - | $81.02 \pm 0.07$ | - | - |
| NDS | - | $68.39 \pm 0.17$ | - | - | $70.17 \pm 0.16$ | - | - | $68.64 \pm 0.21$ | - | - | $78.68 \pm 0.19$ | - | - |
| DS | 2 | $68.63 \pm 0.24$ | 5 | 15 | $70.93 \pm 0.21$ | 4 | 12 | $68.93 \pm 0.23$ | 77 | 314 | $78.79 \pm 0.21$ | 83 | 397 |
| DS | 3 | $68.92 \pm 0.21$ | 7 | 12 | $71.08 \pm 0.17$ | 4 | 11 | $69.12 \pm 0.18$ | 113 | 265 | $78.94 \pm 0.14$ | 137 | 318 |
| DS | 4 | $69.18 \pm 0.19$ | 8 | 9 | $71.16 \pm 0.23$ | 5 | 9 | $69.57 \pm 0.26$ | 169 | 237 | $79.12 \pm 0.23$ | 186 | 253 |
| DS | 5 | $69.77 \pm 0.22$ | 9 | 8 | $71.42 \pm 0.18$ | 8 | 9 | $69.85 \pm 0.19$ | 190 | 218 | $79.51 \pm 0.17$ | 215 | 206 |
| DS | 6 | $69.81 \pm 0.15$ | 9 | 8 | $71.39 \pm 0.22$ | 8 | 8 | $69.71 \pm 0.13$ | 212 | 193 | $80.03 \pm 0.19$ | 268 | 184 |
| DS | 7 | $69.38 \pm 0.18$ | 10 | 6 | $71.34 \pm 0.16$ | 9 | 7 | $70.05 \pm 0.18$ | 295 | 174 | $79.62 \pm 0.23$ | 329 | 161 |
| DS | 8 | $69.12 \pm 0.21$ | 13 | 6 | $71.29 \pm 0.13$ | 11 | 7 | $70.28 \pm 0.27$ | 346 | 138 | $79.93 \pm 0.12$ | 365 | 127 |
| DS | 9 | $69.35 \pm 0.27$ | 18 | 9 | $71.51 \pm 0.23$ | 12 | 9 | $70.52 \pm 0.09$ | 371 | 101 | $80.16 \pm 0.17$ | 397 | 96 |
| DS | 10 | $69.24 \pm 0.19$ | 22 | 11 | $71.16 \pm 0.21$ | 14 | 10 | $70.83 \pm 0.15$ | 408 | 78 | $79.98 \pm 0.12$ | 426 | 78 |
| DS | 20 | $69.26 \pm 0.24$ | 19 | 13 | $71.25 \pm 0.18$ | 14 | 11 | $70.22 \pm 0.21$ | 427 | 77 | $79.41 \pm 0.19$ | 447 | 69 |
| DS | 30 | $69.05 \pm 0.28$ | 14 | 15 | $70.89 \pm 0.25$ | 13 | 13 | $69.48 \pm 0.23$ | 315 | 152 | $79.17 \pm 0.22$ | 342 | 135 |
| DS | 40 | $68.62 \pm 0.25$ | 9 | 16 | $70.56 \pm 0.26$ | 8 | 14 | $69.03 \pm 0.15$ | 223 | 238 | $78.89 \pm 0.20$ | 276 | 244 |

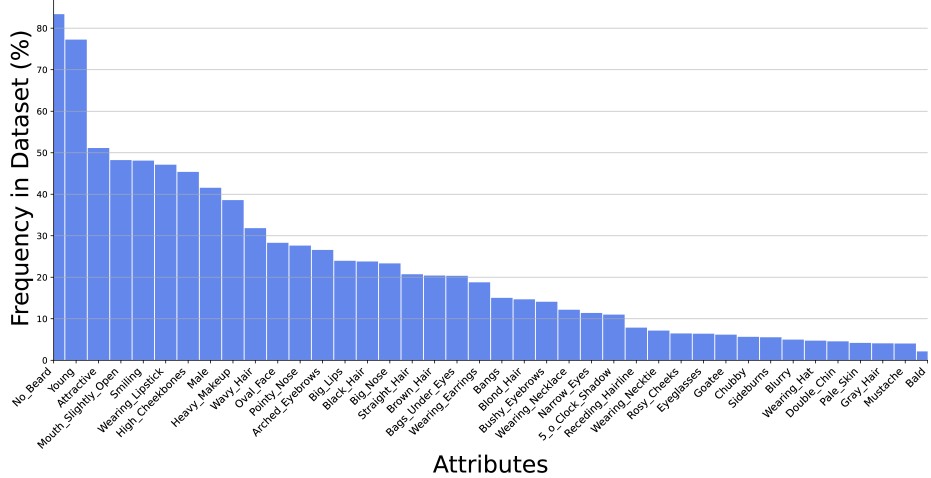

Figure 6: Attribute Distribution in the CelebA Dataset: The histogram depicts the occurrence frequency of each attribute within the images of the dataset. We can see a long-tailed distribution, indicating that some attributes are substantially more prevalent than others.

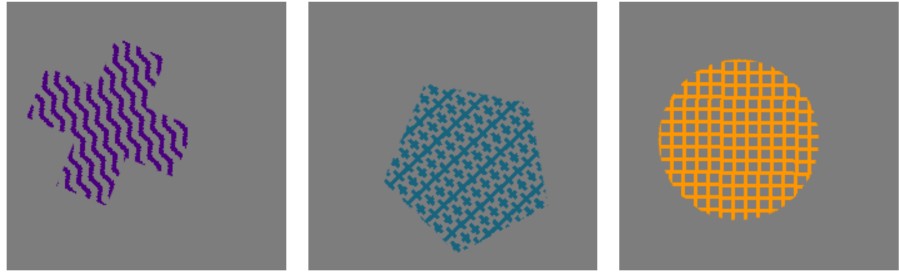

Figure 7: Example images from the Trifeature synthetic dataset (Hermann and Lampinen, 2020).

Table 8: **Test Accuracy and Significant Class Count.** The performance of teacher, Non-Distilled Student (NDS), and Distilled Student (DS) models with a range of temperatures $T$ on the CIFAR-100 dataset with $\alpha = 0.6$ and $\alpha = 0.8$. The number of statistically significantly affected classes comparing the class-wise accuracy of *teacher vs. Distilled Student (DS) models*, denoted #TC, and *Non-Distilled Student (NDS) vs. distilled student models*, denoted #SC.

| Teacher/Student | | CIFAR-100 ( $\alpha = 0.6$) ResNet56/ResNet20 | | | CIFAR-100 ( $\alpha = 0.8$) ResNet56/ResNet20 | | |
|---|---|---|---|---|---|---|---|
| Model | Temp | Test Acc. (%) | #SC | #TC | Test Acc. (%) | #SC | #TC |
| Teacher | - | $70.87 \pm 0.21$ | - | - | $70.87 \pm 0.21$ | - | - |
| NDS | - | $68.39 \pm 0.17$ | - | - | $68.39 \pm 0.17$ | - | - |
| DS | 2 | $68.53 \pm 0.16$ | 8 | 17 | $68.63 \pm 0.24$ | 5 | 15 |
| DS | 3 | $68.65 \pm 0.12$ | 9 | 14 | $68.92 \pm 0.21$ | 7 | 12 |
| DS | 4 | $69.77 \pm 0.20$ | 9 | 12 | $69.18 \pm 0.19$ | 8 | 9 |
| DS | 5 | $69.10 \pm 0.16$ | 10 | 11 | $69.77 \pm 0.22$ | 9 | 8 |
| DS | 6 | $69.42 \pm 0.19$ | 11 | 9 | $69.81 \pm 0.15$ | 9 | 8 |
| DS | 7 | $69.21 \pm 0.15$ | 14 | 7 | $69.38 \pm 0.18$ | 10 | 6 |
| DS | 8 | $69.01 \pm 0.23$ | 16 | 8 | $69.12 \pm 0.21$ | 13 | 6 |
| DS | 9 | $68.89 \pm 0.18$ | 17 | 10 | $69.35 \pm 0.27$ | 18 | 9 |
| DS | 10 | $69.03 \pm 0.21$ | 20 | 10 | $69.24 \pm 0.19$ | 22 | 11 |

Table 9: **Fairness Metrics and Distillation on CelebA.** The performance of teacher, Non-Distilled Student (NDS), and Distilled Student (DS) models with a range of temperatures $T$ on the CelebA dataset. Fairness metrics are presented with regard to the Young demographic attribute for binary Smiling classification and binary Eyeglasses classification. Mean and std. dev. are over five random inits.

| | | CelebA (Smiling) | | | CelebA (Eyeglasses) | | |
|---|---|---|---|---|---|---|---|
| Teacher/Student: | | ResNet-50 / ResNet-18 | | | ResNet-50 / ResNet-18 | | |
| Model | Temp | Test Acc. (%) | EOD | DPD | Test Acc. (%) | EOD | DPD |
| Teacher | – | $93.09 \pm 0.08$ | $\mathbf{4.69 \pm 0.06}$ | $9.41 \pm 0.11$ | $98.57 \pm 0.02$ | $\mathbf{1.18 \pm 0.03}$ | $2.14 \pm 0.08$ |
| NDS | – | $92.03 \pm 0.03$ | $6.11 \pm 0.05$ | $10.60 \pm 0.08$ | $97.51 \pm 0.03$ | $1.76 \pm 0.07$ | $3.21 \pm 0.10$ |
| DS | 1 | $92.12 \pm 0.06$ | $6.02 \pm 0.11$ | $9.97 \pm 0.08$ | $97.62 \pm 0.03$ | $1.69 \pm 0.02$ | $2.73 \pm 0.04$ |
| DS | 2 | $92.14 \pm 0.11$ | $5.67 \pm 0.08$ | $9.75 \pm 0.09$ | $97.66 \pm 0.06$ | $1.65 \pm 0.03$ | $2.59 \pm 0.09$ |
| DS | 3 | $92.53 \pm 0.13$ | $5.45 \pm 0.05$ | $9.63 \pm 0.06$ | $97.78 \pm 0.08$ | $1.62 \pm 0.04$ | $2.48 \pm 0.06$ |
| DS | 4 | $92.17 \pm 0.10$ | $5.36 \pm 0.02$ | $9.38 \pm 0.03$ | $97.95 \pm 0.05$ | $1.57 \pm 0.03$ | $2.41 \pm 0.04$ |
| DS | 5 | $92.29 \pm 0.05$ | $5.39 \pm 0.04$ | $9.30 \pm 0.05$ | $97.83 \pm 0.09$ | $1.55 \pm 0.01$ | $2.33 \pm 0.08$ |
| DS | 6 | $92.26 \pm 0.08$ | $5.30 \pm 0.01$ | $9.38 \pm 0.07$ | $97.78 \pm 0.11$ | $1.56 \pm 0.04$ | $2.27 \pm 0.03$ |
| DS | 7 | $92.12 \pm 0.08$ | $5.26 \pm 0.05$ | $9.17 \pm 0.10$ | $97.82 \pm 0.13$ | $1.53 \pm 0.02$ | $2.23 \pm 0.03$ |
| DS | 8 | $92.66 \pm 0.12$ | $5.22 \pm 0.02$ | $9.05 \pm 0.04$ | $97.71 \pm 0.10$ | $1.50 \pm 0.05$ | $2.19 \pm 0.02$ |
| DS | 9 | $93.18 \pm 0.15$ | $5.14 \pm 0.04$ | $9.01 \pm 0.08$ | $97.86 \pm 0.11$ | $1.46 \pm 0.03$ | $2.16 \pm 0.04$ |
| DS | 10 | $92.57 \pm 0.11$ | $4.98 \pm 0.03$ | $\mathbf{8.86 \pm 0.04}$ | $97.84 \pm 0.12$ | $1.42 \pm 0.04$ | $\mathbf{2.05 \pm 0.09}$ |
| DS | 20 | $92.42 \pm 0.13$ | $5.12 \pm 0.05$ | $9.28 \pm 0.09$ | $97.69 \pm 0.14$ | $1.44 \pm 0.06$ | $2.14 \pm 0.12$ |
| DS | 30 | $92.19 \pm 0.09$ | $5.52 \pm 0.05$ | $9.73 \pm 0.06$ | $97.61 \pm 0.06$ | $1.58 \pm 0.11$ | $2.43 \pm 0.14$ |
| DS | 40 | $92.08 \pm 0.04$ | $6.05 \pm 0.03$ | $10.12 \pm 0.07$ | $97.58 \pm 0.04$ | $1.71 \pm 0.13$ | $2.95 \pm 0.13$ |

Table 10: **Fairness Metrics and Distillation on Trifeature dataset.** The performance of teacher, Non-Distilled Student (NDS), and Distilled Student (DS) models with a range of temperatures $T$ on the Trifeature dataset for shape classification. Fairness metrics are presented with regard to color attributes and texture attributes. Mean and std. dev. are over five random inits.

| | | Trifeature (Color) | | | Trifeature (Texture) | | |
|---|---|---|---|---|---|---|---|
| Teacher/Student: | | ResNet-20 / LeNet-5 | | | ResNet-20 / LeNet-5 | | |
| Model | Temp | Test Acc. (%) | EOD | DPD | Test Acc. (%) | EOD | DPD |
| Teacher | – | $100 \pm 0.00$ | $\mathbf{0.00 \pm 0.00}$ | $3.64 \pm 0.00$ | $94.43 \pm 0.04$ | $\mathbf{1.07 \pm 0.02}$ | $3.76 \pm 0.03$ |
| NDS | – | $90.33 \pm 0.07$ | $2.81 \pm 0.05$ | $4.12 \pm 0.08$ | $86.31 \pm 0.08$ | $2.98 \pm 0.05$ | $4.61 \pm 0.07$ |
| DS | 1 | $92.70 \pm 0.15$ | $1.67 \pm 0.04$ | $3.75 \pm 0.07$ | $87.14 \pm 0.08$ | $2.74 \pm 0.04$ | $4.28 \pm 0.03$ |
| DS | 2 | $92.41 \pm 0.08$ | $1.60 \pm 0.06$ | $3.56 \pm 0.05$ | $87.52 \pm 0.11$ | $2.56 \pm 0.06$ | $4.14 \pm 0.04$ |
| DS | 3 | $92.66 \pm 0.12$ | $1.38 \pm 0.02$ | $3.52 \pm 0.02$ | $87.71 \pm 0.09$ | $2.33 \pm 0.04$ | $4.08 \pm 0.05$ |
| DS | 4 | $92.61 \pm 0.17$ | $1.32 \pm 0.05$ | $3.48 \pm 0.03$ | $87.86 \pm 0.12$ | $2.25 \pm 0.03$ | $3.93 \pm 0.03$ |
| DS | 5 | $93.13 \pm 0.09$ | $1.25 \pm 0.00$ | $3.41 \pm 0.06$ | $88.14 \pm 0.13$ | $2.17 \pm 0.05$ | $3.85 \pm 0.05$ |
| DS | 6 | $93.25 \pm 0.12$ | $1.27 \pm 0.02$ | $3.48 \pm 0.03$ | $88.28 \pm 0.09$ | $2.06 \pm 0.03$ | $3.74 \pm 0.08$ |
| DS | 7 | $92.85 \pm 0.18$ | $1.21 \pm 0.03$ | $3.37 \pm 0.05$ | $88.13 \pm 0.14$ | $1.97 \pm 0.06$ | $3.59 \pm 0.04$ |
| DS | 8 | $92.66 \pm 0.13$ | $1.15 \pm 0.05$ | $3.29 \pm 0.06$ | $88.06 \pm 0.10$ | $1.91 \pm 0.04$ | $3.51 \pm 0.07$ |
| DS | 9 | $93.18 \pm 0.08$ | $1.08 \pm 0.03$ | $3.22 \pm 0.04$ | $88.23 \pm 0.07$ | $1.83 \pm 0.07$ | $3.44 \pm 0.06$ |
| DS | 10 | $92.77 \pm 0.16$ | $1.01 \pm 0.04$ | $\mathbf{3.10 \pm 0.02}$ | $88.33 \pm 0.11$ | $1.77 \pm 0.04$ | $\mathbf{3.37 \pm 0.09}$ |
| DS | 20 | $92.43 \pm 0.12$ | $1.06 \pm 0.06$ | $3.25 \pm 0.06$ | $88.17 \pm 0.10$ | $1.86 \pm 0.08$ | $3.48 \pm 0.07$ |
| DS | 30 | $91.58 \pm 0.08$ | $1.27 \pm 0.08$ | $3.59 \pm 0.07$ | $87.42 \pm 0.14$ | $2.21 \pm 0.10$ | $3.82 \pm 0.13$ |
| DS | 40 | $91.04 \pm 0.13$ | $1.85 \pm 0.09$ | $3.86 \pm 0.07$ | $86.93 \pm 0.17$ | $2.69 \pm 0.12$ | $4.23 \pm 0.15$ |

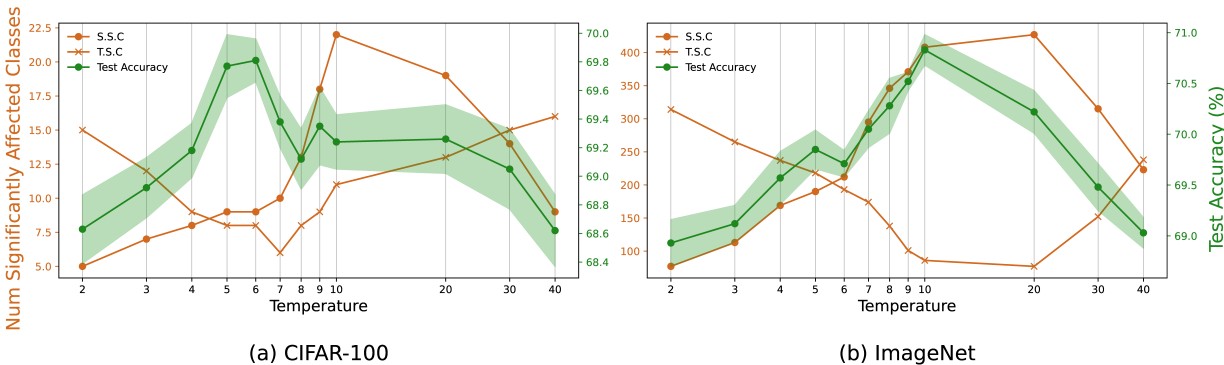

(a) CIFAR-100                    (b) ImageNet

Figure 8: **Temperature vs. Test Accuracy/Class Bias.** Number of non-distilled vs. distilled student significantly affected classes (S.S.C.) and the number of teacher vs. distilled student significantly affected classes (T.S.C.) by distillation in (a) CIFAR-100 (ResNet-56/ResNet-20) and (b) ImageNet datasets (ResNet-50/ResNet-18), with 100 and 1000 total classes respectively. As the temperature used for distillation increases, the S.S.C. rises for both datasets up to a certain T, after which it decreases. Meanwhile, T.S.C. decreases first and then increases. The changes in the distilled student Test Accuracy over all classes are also depicted in the figure.

