# OpenReview forum: "What’s Left After Distillation? How Knowledge Transfer Impacts Fairness and Bias"
_TMLR — Accepted by TMLR_

### Review · Reviewer_ug8H · 2024-12-04

**Summary Of Contributions:**

The following contribution argues that knowledge distillation can influence class-wise accuracies and fairness metrics, even in balanced datasets like CIFAR-100, Tiny ImageNet, and ImageNet.

The authors show how distillation affects up to 41% of classes in terms of accuracy discrepancies between teacher and student models, as well as between distilled and non-distilled student models.

**Audience:**

No

**Broader Impact Concerns:**

Certainly, this study respects the ethical impact very much as it deals with a sensitive and important issue, but it needs an improvement in experimentation as described among the weaknesses.

**Claims And Evidence:**

No

**Requested Changes:**

Based on the weaknesses, I would ask the authors to argue and expand:

- generalisability of results;
- usability in the real world;
- diversification of techniques used andthe  possibility of a more sustained ablation study.

**Strengths And Weaknesses:**

**Strengths:**

- The paper proposes a topical topic and investigates the impact of knowledge distillation on bias and fairness in machine learning models.

- Innovative Approach: Indeed, distillation temperature on bias is an important and significant application.

- Good and comprehensibly introduced results (definitely needs improvement)

**Weaknesses:**

- Lack of Direct Comparisons: There are no direct comparisons with other model compression techniques that do not involve distillation, which is very limiting.

- Limited Depth of Analysis: While the paper explores the impact of temperature on distillation, it lacks a deeper analysis of how different distillation configurations may affect the models differently.

- Generalisability of Results: The results are limited to specific data sets and model configurations is it possible to transfer the findings to other models and scenarios?

- Assessment Methodologies: The methodologies used to assess fairness and bias could benefit from greater diversification to cover a wider range of possible scenarios and less obvious discrimination.

---

> ### Author Response · Authors · 2025-01-09
>
> We sincerely thank the reviewer for their thoughtful and detailed feedback. Below, we clarify the scope of our work and address the specific concerns raised:
>
> - The focus of this study is specifically on the impact of temperature in knowledge distillation on the bias and fairness of the models. As such, direct comparisons with other model compression techniques, which do not involve distillation, we believe fall outside the scope of this work and are not directly relevant to the questions we aim to address.
>
> - We acknowledge that different configurations in knowledge distillation can influence model behavior and outcomes. However, the primary focus of this study is to investigate the specific impact of temperature on bias and fairness.
> That said, we have included experiments with varying alpha values, as summarized in Appendix B (Table 7), to evaluate the generalizability of our findings across different configurations. These results indicate that the trends observed for temperature are consistent across a range of alpha values, reinforcing the robustness of our conclusions.
> While a more exhaustive exploration of additional configurations is beyond the scope of this work, we view this as an important direction for future research and will highlight it as such in the discussion section.
>
> - Our study includes experiments across natural language and computer vision tasks, on a diverse range of datasets (e.g., CIFAR-100, ImageNet, CelebA, Trifeaure and HateXplain) and model architectures (e.g., ResNet-20/ResNet-50, DenseNet, and DistilBERT). This variety was chosen deliberately to evaluate the effects of temperature in knowledge distillation across different modalities and complexities in classification. The consistent trends in fairness improvements observed across these settings suggest that our findings are broadly applicable.
>
> - In addition, the use of knowledge distillation is growing. Our findings provide important insights into how temperature settings in distillation can influence fairness, emphasizing the need for careful consideration of this parameter in sensitive domains such as healthcare and autonomous driving, where fairness and bias can have critical implications.
>
> - In our study, we employed widely recognized fairness metrics such as Demographic Parity Difference (DPD) and Equalized Odds Difference (EOD) for datasets with demographic attributes, while using class-wise performance metrics for balanced datasets like CIFAR-10, CIFAR-100, and ImageNet, which do not inherently exhibit obvious discrimination. The class-wise performance methodology was specifically chosen to identify biases or imbalances at the class level in these datasets. While these methodologies are robust and appropriate for the datasets studied, we acknowledge the importance of exploring additional methodologies to capture subtler or less apparent forms of discrimination.
>
> - To address this, we conducted an additional analysis incorporating individual fairness, which ensures that similar inputs receive similar predictions. This analysis revealed the same overall trend: as the temperature increases, fairness improves, as evidenced by lower individual fairness scores reflecting reduced bias. We believe this additional perspective further strengthens the robustness of our findings and demonstrates our commitment to addressing the reviewer’s concerns.
>
> **Table:** **Individual Fairness Metrics Across Datasets.** Individual fairness scores for Teacher, Non-Distilled Student (NDS), and Distilled Student (DS) models across CelebA, Trifeature, and HateXplain datasets. For DS models, scores are reported for varying temperature values `T`.
>
> | Model | Temp | CelebA (ResNet-50 / ResNet-18) | Trifeature (ResNet-20 / LeNet-5) | HateXplain (Bert-Base / DistilBERT) |
> |-------|------|--------------------------------|----------------------------------|-------------------------------------|
> | Teacher | --   | 0.0407   | 0.016   | 0.0320                              |
> | NDS     | --   | 0.124  | 0.0462  | 0.1078                              |
> | DS      | 1    | 0.113   | 0.0422   | 0.0994                              |
> | DS      | 2    | 0.104     | 0.0407  | 0.0985                              |
> | DS      | 3    | 0.0908   | 0.0393   | 0.0927                              |
> | DS      | 4    | 0.0906    | 0.0387   | 0.0882                              |
> | DS      | 5    | 0.0886   | 0.0384     | 0.0823                              |
> | DS      | 6    | 0.0799      | 0.0377  | 0.0768                              |
> | DS      | 7    | 0.0753    | 0.0356    | 0.0727                              |
> | DS      | 8    | 0.0712    | 0.0349     | 0.0689                              |
> | DS      | 9    | 0.0701    | 0.0341   | 0.0681                              |
> | DS      | 10   | 0.0697       | 0.0338  | 0.0654                              |
>
> Thank you again for your constructive feedback.

---

### Review · Reviewer_NBvF · 2024-12-04

**Summary Of Contributions:**

This submission explores the effects of Knowledge Distillation (KD) on class bias or other fairness metrics when applied to deep neural networks on various datasets. The study reveals significant class-level effects and fairness implications of KD, emphasizing the caution needed when using KD for sensitive application domains.

In particular, the paper provides the following contributions/insights:
*  Distillation significantly alters class-wise accuracy, with up to 41% of classes being affected across datasets. These changes are shown to be non-random.
* The empirical results show temperature-dependent trends, where the number of significantly affected classes increases with higher distillation temperatures when comparing non-distilled vs. distilled student models, whereas it decreases when comparing teacher vs. distilled student models.
* Increasing the distillation temperature generally enhances fairness in the student model, sometimes surpassing even the teacher model’s fairness.
* There are trade-offs between fairness and generalization. Improvements in fairness may not always align with better generalization performance, suggesting that applications of KD should carefully consider trade-offs based on specific use cases.

**Audience:**

Yes

**Claims And Evidence:**

Yes

**Requested Changes:**

I would like the authors to address the concerns raised in the "Weaknesses" section.
First, is my understanding of the contribution accurate? If applicable, could the authors make the necessary clarifications? For instance, does the third bullet point in the Introduction section represent this paper's novel contribution? [This is critical]
Secondly, are there additional justifications or explanations that could enhance the universality and soundness of the observed conclusions (if applicable)? [This would strengthen the work in my view, and also critical.]

**Strengths And Weaknesses:**

Strengths:
* Exploring the effect of knowledge distillation from a fairness perspective is an important and impactful direction with meaningful implications.
* The paper conducts comprehensive experiments across diverse datasets (e.g., ImageNet, CelebA, HateXplain) and models (e.g., ResNets). The scale and complexity of these datasets and models are sufficient to demonstrate meaningful trends and patterns.

Weaknesses:
* This work does not appear to be the first to explore the fairness implications of Knowledge Distillation (KD), as noted in the Related Work section, where Lukasik et al. (2021) and Chai et al. (2022) have previously investigated this direction. If the main contribution of this paper lies in studying the effect of the temperature parameter T, the significance of the contribution feels somewhat diminished.
* Regarding the universality and explainability of the empirical results, there are some concerns. On one hand, the authors seem to observe a trend where higher temperatures (within the range of 2 to 10) improve fairness. On the other hand, as shown in Figure 3 (a) and (b), datasets like CIFAR-100 and ImageNet exhibit somewhat different trends concerning "T.S.C" and "Test Accuracy." This raises questions about the universality of the conclusions. Providing justifications or theoretical explanations for these results would make the findings much more convincing and interesting.

---

> ### Author Response · Authors · 2025-01-09
>
> We sincerely thank the reviewer for their thoughtful feedback, detailed review, and recognition of the contributions of our work.
>
> - While we acknowledge that fairness in KD has been explored in prior work, the primary focus of our study lies in investigating the role of temperature in shaping bias and fairness outcomes.
> Temperature is arguably the most important hyper-parameter in knowledge distillation, as it directly controls the softening of probability distributions during knowledge transfer. Despite its widespread use, temperature is typically tuned solely based on general performance metrics, such as overall accuracy, with little attention paid to its implications for bias and fairness.
> Our work bridges this gap by providing the first systematic exploration of how temperature affects class bias and fairness across various datasets and model architectures. The findings emphasize that temperature not only influences general performance but also has a profound impact on the fairness characteristics of distilled models, highlighting the need to consider fairness alongside traditional metrics when tuning this parameter.
> We believe that this focused analysis on temperature contributes significantly to the understanding of fairness in KD, complementing and extending prior work in this area.
>
> - As discussed in the paper, test accuracy exhibits fluctuations across the temperature range and it has a wide range of variability on datasets. For T.S.C., the trends are consistent up to a temperature of 8. However, the divergence observed for CIFAR-100 beyond this point arises from its increased complexity and fewer instances per class compared to the ImageNet dataset, which contains more samples and richer class diversity.
> The trends shown in Figure 3 primarily illustrate the number of significantly affected classes (T.S.C. and S.S.C.). Importantly, the trend for improving fairness is consistent across all datasets and temperature settings within our defined range. This consistency in fairness trends reinforces universality of our findings.
>
> - The third bullet point in the Introduction highlights how varying the temperature parameter in knowledge distillation can significantly influence class bias and fairness, even when the overall test accuracy remains relatively stable. While prior works have focused on generalization performance or other hyper-parameters like alpha, our work is the first to systematically explore the fairness implications of temperature across multiple datasets and model architectures. This is a key contribution of our study and is supported by empirical evidence demonstrating consistent fairness improvements with increased temperatures.
> Our study includes experiments on a wide range of datasets and model architectures. This diversity helps reinforce the general applicability of our conclusions.
>
> - To further enrich the qualitative analysis, we propose to include the following explanation in the revised manuscript:
> The temperature parameter plays a crucial role in reshaping the output probability distribution during knowledge distillation. At lower temperatures, the probability distribution is sharp, heavily favoring the most likely class, which may overemphasize specific patterns learned by the teacher model. This can amplify biases associated with underrepresented or challenging classes, as the distilled student may rely more on the dominant classes for its generalization.
> Conversely, higher temperatures produce softened probability distributions, which effectively increase the entropy of the knowledge transferred from the teacher to the student. This richer representation allows the student model to learn from secondary patterns in the teacher's predictions, potentially mitigating over-reliance on dominant classes and leading to more equitable treatment of all classes. This mechanism explains why increasing the temperature reduces the number of statistically significantly affected classes and improves fairness metrics in our experiments.
>
> Thank you again for your constructive feedback.

---

### Review · Reviewer_QFKj · 2024-12-29

**Summary Of Contributions:**

The paper investigates the downstream effects of knowledge distillation on class-wise accuracy (class bias) and fairness (w.r.t. sensitive attributes). The authors systematically vary the distillation temperature to assess its impact on the class bias and fairness of distilled student models compared to teacher models and non-distilled student models. Through comprehensive experiments across multiple datasets and various teacher-student architectural configurations, the key contributions are: (1) Demonstrating that knowledge distillation can induce statistically significant class bias in up to 41% of classes, highlighting potential uneven effects when comparing teacher/distilled student and distilled student/non-distilled student models. (2) Establishing that increasing the distillation temperature enhances the fairness of distilled student models, with evidence that, at sufficiently high temperatures, the fairness of the student model can exceed that of the teacher model. This work calls for caution when applying distilled models in sensitive contexts.

**Audience:**

Yes

**Claims And Evidence:**

Yes

**Requested Changes:**

- The title and some sections imply a broad applicability across different hyper-parameters in knowledge distillation, but the results are limited to the effect of temperature—only one among several hyper-parameters. It is recommended that the authors either examine the effects of additional hyper-parameters or revise the title and relevant sections (e.g., the Introduction) to frame the work as a focused case study on temperature.
- It would be interesting to discuss the interplay between $\alpha$ (as defined in Equation 6) and temperature in influencing class bias and model fairness. Such an analysis could offer insights into how these two factors jointly shape the balance between class bias and fairness in student models, especially considering that the hyper-parameters in the knowledge distillation framework are typically tuned jointly.
- While the paper provides sufficient empirical evidence, it would be valuable for the authors to include a more detailed qualitative analysis—perhaps offering some intuition—about why increasing temperature leads to a decrease in the number of statistically significant affected classes or improves fairness. A deeper exploration of the underlying mechanisms could provide readers with a better understanding of the observed phenomena and help clarify how temperature influences class bias and fairness in knowledge distillation.

**Strengths And Weaknesses:**

Strengths:
- The paper highlights an important observation about the class bias and fairness in the context of knowledge distillation, which has implications for both research and application.
- It contains thorough empirical experimentation, using diverse datasets and a range of teacher-student model architectures to support the findings.

Weaknesses:
- The study focuses exclusively on the effect of temperature among the many hyper-parameters in knowledge distillation, leaving the potential interplay and impact of other factors unexplored.
- There is a lack of qualitative discussion to provide insights into why varying temperature has such a profound effect on class bias and fairness, limiting the interpretability of the findings.

---

> ### Author Response · Authors · 2025-01-09
>
> Thank you for your detailed feedback, thoughtful suggestions, and recognition of the contributions of our work. Below, we clarify the scope of our work and address the specific concerns raised:
>
> In the context of knowledge distillation, two primary hyper-parameters are typically considered:
> α, which determines the weighting of the distillation and classification losses in the objective function.
> Temperature, which is used to produce softened probability distributions from logits.
> As highlighted in the related work section, there has been prior research examining the role of α in influencing fairness and bias (e.g., Chai et al., 2022). However, there has been a lack of in-depth studies on the impact of temperature, particularly in relation to class bias and fairness. Consequently, our study focuses on temperature, aiming to address this gap in the literature.
> While our primary focus is temperature, we also experimented with different alpha values to assess the generalizability of our findings. Our results, which are briefly summarized in Appendix B (Table 7), indicate that the observed trends hold across a range of alpha values. This reinforces the robustness of our conclusions regarding the influence of temperature on fairness and bias.
> We acknowledge that the current framing may suggest a broader exploration of hyper-parameters in knowledge distillation.
> Proposed Action: We will revise the title and relevant sections of the paper to explicitly frame this work as a focused analysis on temperature, while referencing related studies on alpha to position our contribution within the broader context of hyper-parameter research.
>
> In addition, we acknowledge the importance of providing interpretability alongside empirical results and have already included some discussion on this topic. For instance:
>
> In Section 5 (Experimental Design and Results), we discuss the relationship between increasing temperature and the decrease in the number of statistically significantly affected classes, highlighting how higher temperatures lead to better alignment of the student model’s biases with those of the teacher.
> In Figure 3, we show how temperature affects the alignment between the teacher and student models, illustrating the nuanced impact of temperature on both class-wise performance and fairness.
> In Section 6 (Discussion), we mention how softened probability distributions at higher temperatures contribute to improved fairness.
>
> While these points provide some intuition, we agree that expanding this discussion will enhance the interpretability of our findings.
> To further enrich the qualitative analysis, we propose to include the following explanation in the revised manuscript:
> The temperature parameter plays a crucial role in reshaping the output probability distribution during knowledge distillation. At lower temperatures, the probability distribution is sharp, heavily favoring the most likely class, which may overemphasize specific patterns learned by the teacher model. This can amplify biases associated with underrepresented or challenging classes, as the distilled student may rely more on the dominant classes for its generalization.
> Conversely, higher temperatures produce softened probability distributions, which effectively increase the entropy of the knowledge transferred from the teacher to the student. This richer representation allows the student model to learn from secondary patterns in the teacher's predictions, potentially mitigating over-reliance on dominant classes and leading to more equitable treatment of all classes. This mechanism explains why increasing the temperature reduces the number of statistically significantly affected classes and improves fairness metrics in our experiments.
>
> Thank you again for your constructive feedback.

---

### Decision · Action_Editor_jduz · 2025-02-05

**Recommendation:** Accept with minor revision

**Comment:**

The main contribution of this work is about how distillation, in particular the temperature parameter, affects fairness metrics. All reviewers found the experimental observations of interest and the question under investigation important and insightful. There was some disagreement on how comprehensive and universal the results are, and concern on the lack of a more in-depth understanding of the observed phenomenon (which the authors partly addressed in the response). In the end we believe the exposed phenomenon is worthwhile for the related fields to be aware of.

Please incorporate the additional experiment on individual fairness in the final revision (and explain the metric for measuring individual fairness, including the similarity measure).

Some other comments:
(1). Intuitively it is clear that as temperature increases, the teacher's soft label tends to uniform. Therefore, with a large alpha and T, the fairness metrics (considered in this work) should tend to 0 while test accuracy potentially also suffers a big drop. Perhaps it is worthwhile to discuss this extreme case and verify it.

(2). The experiments on CIFAR100 and ImageNet have little to do with the fairness metrics, since there is no sensitive attribute on these datasets. At best it shows temperature affects different class in slightly different ways but it has nothing to do with the fairness metrics mentioned in the paper. It'd be great to caution the readers on this point.

(3). I suggest the authors provide more justification behind Eq. (8), in particular the ratio there. For instance, if a temperature results accuracy to drop uniformly by 1% overall all class and all sensitive attributes, should we consider the impact significant and reject the null hypothesis?

(4). Related to above, if a temperature results in the increase of the accuracy of one attribute by 3% while of another by 0% (say, compared to the non-distilled student). Should this be considered unfair? What is a proper baseline to compare to when we discuss the changes of the fairness metrics? Also, how practically relevant are these changes? For instance, in Table 2 (CelebA), is it a big deal when changing DPD from 9.41 to 8.86? Providing some context to interpret and appreciate these changes would be very welcome.

(5). In Table 2, the teacher model achieved 100 accuracy (with 0 std deviation) but its DPD is 3.64. This perhaps suggests the inappropriateness of this metric (when base rates differ much) and I suggest the authors caution the readers about this point.

**Audience:**

Yes, this paper might be of interest to people who work on algorithmic fairness or model distillation.

**Claims And Evidence:**

Yes, the claims are supported with experiments on a variety of datasets and models.